



# MPAS-Seaice (v1.0.0): Sea-ice dynamics on unstructured Voronoi meshes

Adrian K. Turner[1], William H. Lipscomb[2], Elizabeth C. Hunke[1], Douglas W. Jacobsen[4], Nicole Jeffery[3], Darren Engwirda[1], Todd D. Ringler[5], and Jonathan D. Wolfe[1]

[1]Theoretical Division, Los Alamos National Laboratory, Los Alamos, NM, USA
[2]Climate and Global Dynamics Laboratory, National Center for Atmospheric Research, Boulder, CO, USA
[3]Computer, Computational, and Statistical Sciences Division, Los Alamos National Laboratory, Los Alamos, NM, USA
[4]formerly Theoretical Division, Los Alamos National Laboratory, Los Alamos, NM, USA
[5]Science Technology Policy Institute, Washington, D.C. 20006

**Correspondence:** Adrian K. Turner (akt@lanl.gov)

**Abstract.** We present MPAS-Seaice, a sea-ice model which uses the Model for Prediction Across Scales (MPAS) framework and Spherical Centroidal Voronoi Tessellation (SCVT) unstructured meshes. As well as SCVT meshes, MPAS-Seaice can run on the traditional quadrilateral grids used by sea-ice models such as CICE. The MPAS-Seaice velocity solver uses the Elastic-Viscous-Plastic (EVP) rheology, and the variational discretization of the internal stress divergence operator used by CICE, but
adapted for the polygonal cells of MPAS meshes, or alternatively an integral ("weak") formulation of the stress divergence operator. An incremental remapping advection scheme is used for mass and tracer transport. We validate these formulations with idealized test cases, both planar and on the sphere. The variational scheme displays lower errors than the weak formulation for the strain rate operator but higher errors for the stress divergence operator. The variational stress divergence operator displays increased errors around the pentagonal cells of a quasi-uniform mesh, which is ameliorated with an alternate formu-
lation for the operator. MPAS-Seaice shares the sophisticated column physics and biogeochemistry of CICE, and when used with quadrilateral meshes can reproduce the results of CICE. We have used global simulations with realistic forcing to validate MPAS-Seaice against similar simulations with CICE and against observations. We find very similar results compared to CICE with differences explained by minor differences in implementation such as with interpolation between the primary and dual meshes at coastlines. We have assessed the computational performance of the model, which, because it is unstructured, runs
70% as fast as CICE for a comparison quadrilateral simulation. The SCVT meshes used by MPAS-Seaice allow culling of equatorial model cells and flexibility in domain decomposition, improving model performance. MPAS-Seaice is the current sea-ice component of the Energy Exascale Earth System Model (E3SM).

## 1 Introduction

Sea ice, the frozen surface of the sea at high latitudes, is an important component of the Earth climate system. Rejection of salt
during sea-ice formation helps drive the thermohaline circulation (Killworth, 1983), and its high reflectivity increases planetary albedo and can help drive the polar amplification of climate change through an albedo feedback mechanism (Ingram et al.,





1989). Numerical modeling of sea-ice dynamics and thermodynamics is an important tool in understanding global climate. One of the most popular sea-ice models currently in use is CICE (Hunke et al., 2015). CICE approximates the sea-ice cover as a continuous fluid, and uses an Elastic-Viscous-Plastic (EVP) rheology to describe the relationship between stress and strain

in that fluid (Hunke and Dukowicz, 1997). This rheology adds a numerical elasticity to the Viscous-Plastic rheology of Hibler (1979) to allow explicit time-stepping and parallelization of the algorithm. CICE uses a quadrilateral structured mesh, and has been used with both displaced-pole (Smith et al., 1995) and tripolar (Murray, 1996) grids. The strain rate and internal ice stress divergence operators used by CICE are based on a variational principle (Hunke and Dukowicz, 1997) and account for metric effects in the curvilinear coordinates (Hunke and Dukowicz, 2002).

Recently, unstructured mesh climate models have been gaining popularity (e.g. Ringler et al., 2013; Wang et al., 2014). While all the interior vertices of a structured mesh enjoy the same topological connectivity, the vertices of an unstructured mesh have arbitrary topological connectivity (Bern and Plassmann, 2000). This allows unstructured meshes to concentrate their model degrees of freedom in regions of interest, improving computational efficiency, while avoiding the difficulties of open boundary conditions in limited-domain regional models. The cost of this flexibility in mesh adaptivity, however, is that data

access is less straightforward and requires an explicit accounting of the mesh connectivity. Several unstructured sea-ice models have been developed. Hutchings et al. (2004) implemented and demonstrated a finite-volume, cell-centered discretization. The Unstructured-Grid CICE (UG-CICE; Gao et al., 2011) model, which is built on the FVCOM framework (Chen et al., 2009), also uses a finite-volume formulation. In contrast, finite-element discretizations have been implemented by Lietaer et al. (2008), by Danilov et al. (2015) in the Finite-Element Sea Ice Model (FESIM), and by Mehlmann and Korn (2021) in the

sea-ice component of the Icosahedral Nonhydrostatic Weather and Climate Model (ICON). UG-CICE, FESIM, and ICON all use triangular elements in their meshes.

The Model for Prediction Across Scales (MPAS) framework is another recently developed unstructured modeling framework, which uses a spherical centroidal Voronoi tessellation to form a mesh (Ringler et al., 2010). Several climate model components have been built with the MPAS modeling framework, including ocean (MPAS-O; Ringler et al., 2013), atmo-

sphere (MPAS-A; Skamarock et al., 2012) and land ice (MALI; Hoffman et al., 2018) models. Here, we describe a new MPAS model for sea ice: MPAS-Seaice. This model uses a 'B' Arakawa type grid (Arakawa and Lamb, 1977) with sea-ice velocity defined on cell vertices and tracers defined at cell centers. Grid cells in MPAS meshes are polygons with four or more sides, rather than triangles. This allows MPAS-Seaice to use the same mesh as either structured quadrilateral models such as CICE, or match the mesh used by unstructured ocean models such as MPAS-Ocean. Matching the ocean mesh is required for coupling

sea-ice components within some global climate models such as CESM (Danabasoglu et al., 2020) and E3SM (Golaz et al., 2019), and simplifies the ocean/sea-ice coupling methodology.

We have implemented two discretizations of the momentum equation for MPAS-Seaice. The first is based on the variational scheme used by CICE (Hunke and Dukowicz, 2002). The second uses the integral ("weak") form of the relevant operators. We implement this second operator method as a comparison to help investigate sources of error in the standard variational scheme.

Within the variational scheme we also investigate several enhancements with the aim of reducing errors associated with the variational scheme. MPAS-Seaice uses the same EVP rheology and column physics as CICE. Sea-ice tracer transport uses an





incremental remapping scheme (Lipscomb and Hunke, 2004; Lipscomb and Ringler, 2005) modified for the MPAS mesh. In section 2, we describe the modeling approach used in MPAS-Seaice, focusing on the solution of the sea-ice momentum equation and tracer transport. In section 3 we validate this new model, both with idealized test cases and with global simulations. In

section 4 we consider the computational performance of the model, and conclude in section 5. We focus in this paper on simulations with quasi-uniform global meshes: variable resolution meshes will be considered in later publications.

## 2 Model description

### 2.1 The MPAS framework

The MPAS mesh uses a Spherical Centroidal Voronoi Tessellation (SCVT), as described in Ringler et al. (2010), and consists

of a primary mesh of the Voronoi cells tessellated on the sphere and a dual triangular Delaunay mesh, formed from joining the Voronoi cell centers into a triangulation. Figure 1 shows a schematic of part of a MPAS mesh with the primary mesh shown as a solid line and the dual mesh shown as a dashed line. The primary mesh cells have their Voronoi generating points coincident with the centroid of the cell. The mesh consists of three types of points arranged on the sphere: the Voronoi cell center points, vertex points of the Voronoi cell, and edge points at the midpoint of the Voronoi cell edges (see Fig. 1). The mesh is constructed

so that a line joining neighboring cell centers is perpendicular to the edge that line passes through. On a typical quasi-uniform MPAS mesh, the majority of the cells are hexagons, but at least twelve pentagons are needed to complete the tessellation. In general, the cells are not regular polygons and may consist of polygons with edge numbers greater than or equal to four. As well as quasi-uniform grids, meshes can be generated with regions of enhanced resolution, allowing computational effort to be focused in regions of interest. The MPAS mesh standard can also represent the quadrilateral meshes used by CICE, where four

instead of three edges meet at each vertex. With these quadrilateral meshes the dual mesh is also quadrilateral.

### 2.2 Velocity solver

MPAS-Seaice uses a 'B' Arakawa type grid (Arakawa and Lamb, 1977) with both components of velocity defined at cell vertices, and with sea-ice concentration, volume and other tracers defined at cell centers (see Fig. 1). When using CICE-like quadrilateral meshes, the velocity solver algorithm of MPAS-Seaice reduces to that of CICE, allowing CICE and MPAS-Seaice

to use identical test cases and supporting rapid testing and development of MPAS-Seaice.

In CICE the velocity components are aligned with the quadrilateral mesh. This is not possible, in general, with MPAS-Seaice since a SCVT MPAS mesh does not have edges with perpendicular directions as in a quadrilateral mesh. Instead, the velocity components at a given MPAS vertex are defined as eastwards ($u$) and northwards ($v$), irrespective of the orientation of edges joining that vertex. Such a definition, however, would result in a convergence of $v$ components of velocity at the geographic

poles and strong metric terms in the velocity solution. Consequently, we rotate $u$ and $v$ so that their pole lies on the geographical equator at $0°$ longitude. The polar regions then have the smallest metric effects on the globe.



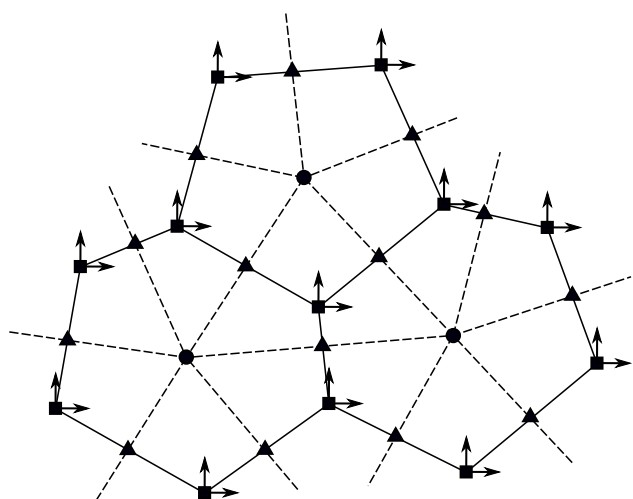

**Figure 1.** Schematic representation of three cells in an example MPAS mesh. The mesh is composed of cell centers (*circles*), cell edge points (*triangles*), and cell vertices (*squares*). The dual triangular Delaunay mesh is formed by joining cell centers (*dashed line*). MPAS-Seaice uses a 'B' type grid, with both velocity components defined at cell vertices. Unlike traditional quadrilateral meshes, the directions of the velocity components in MPAS-Seaice are not in general aligned with the cell edges.

To prognose sea-ice velocity we solve the same sea-ice momentum equation as CICE (Hibler, 1979; Hunke and Dukowicz, 1997):

$$m\frac{\partial \boldsymbol{u}}{\partial t} = \boldsymbol{\nabla} \cdot \boldsymbol{\sigma} + \boldsymbol{\tau_a} + \boldsymbol{\tau_w} - \hat{\boldsymbol{k}} \times mf\boldsymbol{u} - mg\boldsymbol{\nabla}H_o. \tag{1}$$

Here $m$ is the mass of snow and ice per unit area, $\boldsymbol{u}$ is the sea-ice velocity, $\boldsymbol{\sigma}$ is the ice internal stress tensor, $\boldsymbol{\tau_a}$ and $\boldsymbol{\tau_w}$ are the horizontal stresses due to atmospheric winds and ocean currents respectively, $\hat{\boldsymbol{k}}$ is the unit vector normal to the Earth surface, $f$ is the Coriolis parameter, $g$ is the acceleration due to gravity and $H_o$ is the ocean surface height. The second to last term represents the Coriolis force and the last term represents the force due to the ocean surface tilt. Only the internal stress divergence and ocean surface tilt terms depend on horizontal differential operators. During coupled simulations the

ocean model provides the ocean surface tilt term, whereas in non-coupled simulations we assume that the ocean currents are in geostrophic balance so that

$$mg\boldsymbol{\nabla}H_o = mf\hat{\boldsymbol{k}} \times \boldsymbol{u_o} \tag{2}$$

where $\boldsymbol{u_o}$ is the geostrophic component of the ocean surface velocity. Consequently, only the internal stress divergence depends on the properties of the horizontal grid, and only adaptations to this stress term are required to adapt the velocity solver of CICE

to MPAS meshes. The other terms in the momentum equation are solved in an identical way to CICE.

Determination of the divergence of the internal stress can be broken down into three stages:

1. Determine the strain rate tensor from the velocity field.





2. Determine the stress tensor at a point, through a constitutive relation, from the strain rate tensor at that point.

3. Calculate the divergence of this stress tensor.

As in CICE we use an Elastic-Viscous-Plastic (EVP) rheology (Hunke and Dukowicz, 1997) for the constitutive relation. This step does not depend on the details of the horizontal mesh and we use the same formulation as CICE. We develop two schemes to calculate the strain rate tensor and the divergence of internal stress on MPAS meshes. A variational scheme is based on that used in CICE (Hunke and Dukowicz, 2002), whereas a weak scheme uses the line-integral forms of the symmetric gradient and divergence operators. These schemes are described in the following sections.

### 2.2.1 Variational Scheme

We develop a variational scheme for calculating the stress divergence based on that of Hunke and Dukowicz (2002) but adapted for arbitrarily shaped and sided convex polygons. The principal change needed to adapt Hunke and Dukowicz (2002) to polygonal cells is a generalization of the basis functions from bilinear to a basis compatible with polygonal cells. Here, the directions of the $u$ and $v$ components of the velocity do not follow grid directions as in Hunke and Dukowicz (2002). The 

variational scheme is based on the fact that over the entire domain, $\Omega$, and ignoring boundary effects, the total work done by the internal stress is equal to the dissipation of mechanical energy:

$$\int_\Omega \boldsymbol{u} \cdot (\boldsymbol{\nabla} \cdot \boldsymbol{\sigma}) \mathrm{d}A = -\int_\Omega (\sigma_{11}\dot{\epsilon}_{11} + 2\sigma_{12}\dot{\epsilon}_{12} + \sigma_{22}\dot{\epsilon}_{22}) \mathrm{d}A. \tag{3}$$

Here $\dot{\boldsymbol{\epsilon}}$ is the strain rate tensor and the integrals are area integrals over the whole model domain. The work done in the whole domain can be split into a sum over the contribution to the work done in each cell on the dual Delaunay mesh. These dual cells 

consists of either triangles (for SCVT meshes), or quadrilaterals (for quadrilateral meshes) surrounding a single primary mesh vertex point where the discretized velocity is defined. Equation (3) can then be written as

$$\sum_i^{n_d} \int_i \boldsymbol{u} \cdot (\boldsymbol{\nabla} \cdot \boldsymbol{\sigma}) \mathrm{d}A = D(u_1, u_2, ..., u_n, v_1, v_2, ..., v_{n_d}) \tag{4}$$

where the left-side sum is over the $n_d$ cells of the dual mesh, the integral is an area integral over each dual cell, and the dissipation of mechanical energy has been written as a function of the discretized velocity components. Writing the two 

components of the stress divergence as $F_u = (\nabla \cdot \sigma)_u$ and $F_v = (\nabla \cdot \sigma)_v$,

$$\sum_i^{n_d} \int_i (uF_u + vF_v) \mathrm{d}A = D(u_1, u_2, ..., u_n, v_1, v_2, ..., v_{n_d}). \tag{5}$$

The simplest assumption for the velocity and stress divergence components for the left hand side of Eq. (5) is that these quantities are constant within the dual cell. With this assumption it follows that

$$\sum_i^{n_d} (u_i F_{ui} + v_i F_{vi}) A_{ui} = D(u_1, u_2, ..., u_n, v_1, v_2, ..., v_{n_d}), \tag{6}$$





where $A_{ui}$ is the area of the dual mesh cell. The variation of these expressions with respect to the $u$ component of the discretized velocity at a particular vertex point $j$ is given by

$$\frac{\partial}{\partial u_j} \sum_i^{n_d} (u_i F_{ui} + v_i F_{vi}) A_{ui} = \frac{\partial}{\partial u_j} D(u_1, u_2, ..., u_n, v_1, v_2, ..., v_{n_d}). \tag{7}$$

As discussed in Hunke and Dukowicz (2002), stress components are assumed not to be functions of velocity for this variational calculation, so,

$$F_{uj} = \frac{1}{A_{uj}} \frac{\partial}{\partial u_j} D(u_1, u_2, ..., u_n, v_1, v_2, ..., v_{n_d}). \tag{8}$$

$F_v$ is obtained in a similar way by taking the variation of $D$ with respect to $v_j$. The dissipation of mechanical energy, $D$, can be split into three terms:

$$D = D_1 + D_2 + D_3 \tag{9}$$

with

$$D_1 = -\int \sigma_{11} \dot{\epsilon}_{11} dA, \quad D_2 = -\int 2\sigma_{12} \dot{\epsilon}_{12} dA, \quad D_3 = -\int \sigma_{22} \dot{\epsilon}_{22} dA. \tag{10}$$

We will calculate the contribution to $F_u$ and $F_v$ from $D_1$. Similar contributions come from $D_2$ and $D_3$. Using the expression for $\dot{\epsilon}_{11}$ in terms of the velocity components and latitude $\theta$, $D_1$ becomes

$$D_1 = -\int \sigma_{11} \left[ \frac{\partial u}{\partial x} - \frac{v \tan \theta}{r} \right] dA \tag{11}$$

where $x$ and $y$ are locally Cartesian coordinates, with $x$ in the rotated eastwards direction and $y$ in the rotated northwards direction, and $r$ is the radius of the Earth. The second term in $\dot{\epsilon}$ accounts for the metric effects of the curved domain (Batchelor, 1967). The integral can be broken up into a sum over the $n_p$ cells in the primary mesh:

$$D_1 = -\sum_k^{n_p} \int_k \sigma_{11} \left[ \frac{\partial u}{\partial x} - \frac{v \tan \theta}{r} \right] dA \tag{12}$$

where the integral is over the interior area of the $k$th cell. To perform this integral we use a set of basis functions, $\mathcal{W}_l$, to represent functions within a cell of the primary mesh. These basis functions are such that if a function, $\psi$, has a value of $\psi_l$ at vertex $l$ of a cell, then the value of the function at a position $(x, y)$ within the cell can be approximated as

$$\psi(x, y) = \sum_l^{n_v} \psi_l \mathcal{W}_l(x, y) \tag{13}$$

where the sum is over the $n_v$ vertices of the cell in the primary mesh. Necessary properties for these basis functions are that

$$\sum_l^{n_v} \mathcal{W}_l(x, y) = 1 \tag{14}$$





across the cell, and that

$$
\mathcal{W}_l(x,y)
\begin{cases}
1 & \text{if } (x,y) \text{ at vertex } l \\
0 & \text{if } (x,y) \text{ at any other vertex.}
\end{cases}
\tag{15}
$$

We provide two options for the choice of basis functions, $\mathcal{W}_l$: Wachspress basis functions and Piecewise Linear (PWL) basis functions. Both basis functions have a value of one on vertex $l$ and zero on the other vertices of a cell, and are linear on the cell boundaries. The Wachspress basis function for the $i$th vertex of a polygon with $n$ sides is given by (Dasgupta, 2003)

$$
\mathcal{W}_i = \frac{\mathcal{N}_i}{\sum_j^n \mathcal{N}_j}
\tag{16}
$$

where

$$
\mathcal{N}_i(x,y) = \kappa_i \prod_{j \neq i, j \neq i+1}^{j=n} l_j(x,y)
\tag{17}
$$

and

$$
\kappa_i = \kappa_{i-1} \left( \frac{a_{i+1}(x_{i-1} - x_i) + b_{i+1}(y_{i-1} - y_i)}{a_{i-1}(x_i - x_{i-1}) + b_{i-1}(y_i - y_{i-1})} \right); \kappa_1 = 1.
\tag{18}
$$

$l_j(x,y)$ is the line equation for the $j$ polygon edge such that

$$
l_j(x,y) = 1 - a_j x - b_j y = 0.
\tag{19}
$$

When written out, $\mathcal{W}_i$ becomes a rational polynomial of the form

$$
\mathcal{W}_i^{(n)}(x,y) = \frac{\mathcal{P}^{(n-2)}(x,y)}{\mathcal{P}^{(n-3)}(x,y)}
\tag{20}
$$

where $\mathcal{P}^{(m)}(x,y)$ is a $m$-degree polynomial in $x,y$. The integrals of the Wachspress basis function within a cell are performed using the eighth order quadrature rules of Dunavant (1985). For quadrilateral meshes the Wachspress basis functions reduce to the bilinear basis functions used in CICE. For the four vertices of the quadrilateral cells, these are given by

$$
w_1 = (1 - \xi_1)(1 - \xi_2)
$$
$$
w_2 = (1 - \xi_1)\xi_2
$$
$$
w_3 = \xi_1(1 - \xi_2)
$$
$$
w_4 = \xi_1 \xi_2
$$

where $\xi_1$ and $\xi_2$ are the transformed unit square coordinates of the cell (Hunke, 2001).

PWL basis functions divide the polygonal cell into sub-triangles and use a linear basis within each sub-triangle (Bailey et al., 2008). To divide the polygonal cell into sub-triangles, a point is chosen within the cell and sub-triangles formed using this point and two adjacent vertices. The central point in the cell, $\mathbf{x}_c$, is chosen as

$$
\mathbf{x}_c = \sum_i^{n_v} \alpha_i \mathbf{x}_i
\tag{21}
$$





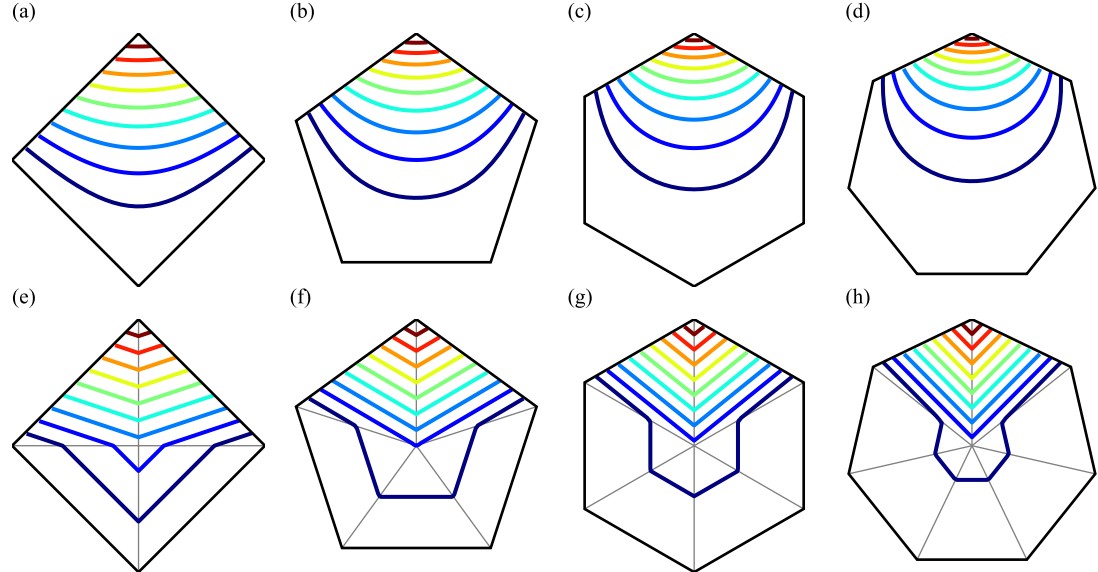

**Figure 2.** Example basis functions for the (a-d) Wachspress and (e-h) PWL methods. The basis function is shown for the top vertex for a (a,e) square, (b,f) pentagonal, (c,g) hexagonal, and (d,h) heptagonal cell. Contour levels are drawn between 0.1 (dark blue) and 0.9 (red) at 0.1 intervals.

where the sum is over the $n_v$ vertices of the cell each with position $\mathbf{x}_i$. The simplest choice for the $\alpha_i$ is to set them all equal to the inverse of the number of cell vertices, $1/n_v$. Example basis functions for the Wachspress and PWL methods are shown in Fig. 2.

Using those basis functions to expand $\sigma_{11}$ (with basis index $l$), $u$, and $v$ (with basis index $m$), Eq. (12) can be written as

$$D_1 = -\sum_k^{n_p} \int_k \left[ \sum_l^{n_v} \sigma_{11l}\mathcal{W}_l \cdot \sum_m^{n_v} \left( u_m \frac{\partial \mathcal{W}_m}{\partial x} - \frac{\tan\theta}{r} v_m \mathcal{W}_m \right) \right] \mathrm{d}A \tag{22}$$

where the derivative with respect to $x$ has been taken inside the summation. Rearranging, this becomes

$$D_1 = -\sum_k^{n_p} \sum_l^{n_v} \sum_m^{n_v} \sigma_{11l} \left( u_m \int_k \mathcal{W}_l \frac{\partial \mathcal{W}_m}{\partial x} \mathrm{d}A - \frac{\tan\theta}{r} v_m \int_k \mathcal{W}_l \mathcal{W}_m \mathrm{d}A \right). \tag{23}$$

In moving the integral, we have assumed that $\theta$, the latitude, is constant in the cell. The terms involving integrals are now only a function of the geometry of the mesh and can be calculated once during the initialization phase of the model run. Defining

$$\mathcal{S}_{lm}^x = \int_k \mathcal{W}_l \frac{\partial \mathcal{W}_m}{\partial x} \mathrm{d}A \tag{24}$$

and

$$\mathcal{T}_{lm} = \int_k \mathcal{W}_l \mathcal{W}_m \mathrm{d}A. \tag{25}$$



we have

$$D_1 = -\sum_k^{n_p} \sum_l^{n_v} \sum_m^{n_v} \sigma_{11l} \left( u_m \mathcal{S}_{lm}^x - \frac{\tan\theta}{r} v_m \mathcal{T}_{lm} \right). \tag{26}$$

Taking the variation with respect to a discretized velocity component at a particular vertex point, $j$, as in Eq. (8), now gives us the contribution from $D_1$ to the components of the stress divergence tensor at that velocity point:

$$(\nabla \cdot \sigma)_{u_j}^{D_1} = \frac{\delta D_1}{\delta u_j} = -\sum_k^{n_p} \sum_l^{n_v} \sigma_{11l} \mathcal{S}_{lj}^x$$

$$(\nabla \cdot \sigma)_{v_j}^{D_1} = \frac{\delta D_1}{\delta v_j} = \sum_k^{n_p} \sum_l^{n_v} \sigma_{11l} \frac{\tan\theta}{r} \mathcal{T}_{lj} \tag{27}$$

Only cells that border the vertex point $j$ contribute to the $k$ sum over cells. The total stress divergence at the point $j$ is then the sum of the contributions from $D_1$, $D_2$, and $D_3$:

$$(\nabla \cdot \sigma)_{u_j} = (\nabla \cdot \sigma)_{u_j}^{D_1} + (\nabla \cdot \sigma)_{u_j}^{D_2} + (\nabla \cdot \sigma)_{u_j}^{D_3}$$

$$(\nabla \cdot \sigma)_{v_j} = (\nabla \cdot \sigma)_{v_j}^{D_1} + (\nabla \cdot \sigma)_{v_j}^{D_2} + (\nabla \cdot \sigma)_{v_j}^{D_3}. \tag{28}$$

All that remains is to determine the stress for each cell at its vertices. As in the formulation in CICE, each cell has its own stress value at its vertices, so each vertex has several values of the stress, each corresponding to a different surrounding cell. The stresses are calculated from the strain rate tensor at each vertex using the constitutive relation (see Section 2a of Hunke and Dukowicz, 2002). Including metric effects (Batchelor, 1967), the strain rate tensor is given by

$$\dot\epsilon_{11} = \frac{\partial u}{\partial x} - \frac{v\tan\theta}{r}$$

$$\dot\epsilon_{22} = \frac{\partial v}{\partial y}$$

$$\dot\epsilon_{12} = \frac{1}{2}\left( \frac{\partial u}{\partial y} + \frac{\partial v}{\partial x} \right) + \frac{u\tan\theta}{2r}. \tag{29}$$

The strain rate tensor at cell vertex $l$ is then given by

$$\dot\epsilon_{11l} = \sum_m^{n_v} u_m \left.\frac{\partial \mathcal{W}_m}{\partial x}\right|_l - \frac{v_l \tan\theta_l}{r}$$

$$\dot\epsilon_{22l} = \sum_m^{n_v} v_m \left.\frac{\partial \mathcal{W}_m}{\partial y}\right|_l$$

$$\dot\epsilon_{12l} = \frac{1}{2}\left( \sum_m^{n_v} u_m \left.\frac{\partial \mathcal{W}_m}{\partial y}\right|_l + \sum_m^{n_v} v_m \left.\frac{\partial \mathcal{W}_m}{\partial x}\right|_l \right) + \frac{u_l \tan\theta_l}{2r}. \tag{30}$$

The derivatives of the basis functions are taken at cell vertex $l$.

One issue with the above derivation is the ambiguity of defining the correct dual cell area, $A_{u_j}$, for Eq. (8) since the cell center positions do not appear anywhere else in the formulation. In section 3.1.2, in a unit sphere test case with analytical





input stress fields, it is evident that using the default MPAS dual cell area for $A_{uj}$ results in large errors in the calculated stress divergence field around the 12 pentagonal cells found on a quasi-uniform SCVT mesh. An alternative to using the MPAS dual cell area is to assume that the velocity and divergence are given by the same basis functions as used for the dissipation of mechanical energy. Then

$$\sum_i^{n_d} \int_i (uF_u + vF_v)\mathrm{d}A = \sum_i^{n_d}\left[\int_i \sum_l^{n_v} u_l \mathcal{W}_l \sum_m^{n_v} F_{um}\mathcal{W}_m + \int_i \sum_o^{n_v} v_o \mathcal{W}_o \sum_q^{n_v} F_{vq}\mathcal{W}_q\right]\mathrm{d}A. \tag{31}$$

Taking the variation with respect to the discretized velocity component $u_j$

$$\sum_i^{n_d}\sum_m^{n_v} F_{um} \int_i \mathcal{W}_j \mathcal{W}_m \mathrm{d}A = \frac{\partial}{\partial u_j} D(u_1, u_2, ..., u_n, v_1, v_2, ..., v_{n_d}), \tag{32}$$

where the sum over $i$ cells is reduced to those surrounding the $j$ vertex. This generates an undesirable implicit problem necessitating a global matrix inversion, which can be avoided if the stress divergence varies slowly spatially. Then

$$F_{uj}\sum_i^{n_d}\int_i \sum_m^{n_v} \mathcal{W}_j \mathcal{W}_m \mathrm{d}A \simeq \frac{\partial}{\partial u_j} D(u_1, u_2, ..., u_n, v_1, v_2, ..., v_{n_d}) \tag{33}$$

where the sum over $m$ is over the vertices in each cell surrounding the $j$ vertex. This suggests an alternative to $A_{uj}$ given by

$$A'_{uj} = \int_i \sum_m^{n_v} \mathcal{W}_j \mathcal{W}_m \mathrm{d}A. \tag{34}$$

We find that approximating $A_{uj}$ with $A'_{uj}$ reduces the error in the stress divergence operator as compared to using the standard dual mesh areas.

### 2.2.2 Weak Scheme

An alternative method of deriving operators is with an integral method where the divergence theorem is used to equate the integral form of the operator to a flux integrated around a closed loop. This results in a "weak" method, as opposed to "strong" methods, such as finite difference schemes, where differential operators are directly approximated. One potential advantage of the weak scheme is that it solves the conservative form of the momentum equation and can handle non-smooth solution features (such as sharp fronts) consistently.

For the weak scheme we use line integrals around cells in the primary and dual meshes to calculate the strain rate tensor and the stress divergence, respectively. To determine the strain rate tensor we start from the following vector identity:

$$\boldsymbol{\nabla}\cdot(\boldsymbol{u}\otimes\boldsymbol{v}) = (\boldsymbol{u}\cdot\boldsymbol{\nabla})\boldsymbol{v} + (\boldsymbol{\nabla}\cdot\boldsymbol{u})\boldsymbol{v} \tag{35}$$

and from the divergence theorem:

$$\int_\Omega [\boldsymbol{\nabla}\cdot(\boldsymbol{u}\otimes\boldsymbol{v})]\,\partial\Omega = \oint_S [\boldsymbol{n}\cdot(\boldsymbol{u}\otimes\boldsymbol{v})]\,\partial S = \oint_S [(\boldsymbol{n}\cdot\boldsymbol{u})\boldsymbol{v}]\,\partial S \tag{36}$$



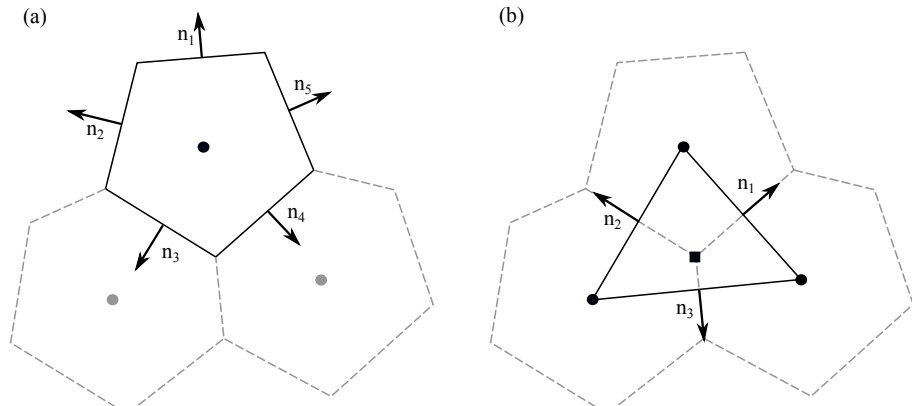

**Figure 3.** Contour integration lines used by the weak stress divergence operator scheme. (a): Strain rate at cell centers (*circle*) are calculated from line integrals around primary mesh cells (*solid line*). (b): Stress divergence at cell vertices (*square*) are calculated from line integrals around the dual mesh cells (*solid line*). Directions of normal vectors used in the integrals are shown for both figures.

where $\boldsymbol{n}$ is a normal vector to the closed surface $S$ and $\otimes$ is the tensor product. If Eq. (35) is integrated over $\Omega$, using Eq. (36) we obtain

$$\int_{\Omega} [(\boldsymbol{u} \cdot \boldsymbol{\nabla})\boldsymbol{v} + (\boldsymbol{\nabla} \cdot \boldsymbol{u})\boldsymbol{v}] \, \partial\Omega = \oint_{S} [(\boldsymbol{n} \cdot \boldsymbol{u})\boldsymbol{v}] \, \partial S \tag{37}$$

If $\boldsymbol{u}$ is chosen as constant then $\boldsymbol{\nabla} \cdot \boldsymbol{u}$ vanishes as does the second term of the left-hand side of Eq. (37). Taking, also, $\boldsymbol{u}$ sequentially as the cartesian unit vectors spanning $\Omega$ and summing the results we obtain

$$\int_{\Omega} [\boldsymbol{\nabla}\boldsymbol{v}] \, \partial\Omega = \oint_{S} [\boldsymbol{n} \otimes \boldsymbol{v}] \, \partial S. \tag{38}$$

The symmetric version of this operator is then obtained as:

$$\int_{\Omega} [\boldsymbol{\nabla}_{\boldsymbol{S}}\boldsymbol{v}] \, \partial\Omega = \oint_{S} \frac{1}{2} [\boldsymbol{n} \otimes \boldsymbol{v} + \boldsymbol{v} \otimes \boldsymbol{n}] \, \partial S \tag{39}$$

The strain rate at a point is then obtained from the limit

$$\dot{\boldsymbol{\epsilon}} = \boldsymbol{\nabla}_S \boldsymbol{v} = \lim_{A \to 0} \frac{1}{A} \oint_{S} \frac{1}{2} [\boldsymbol{n} \otimes \boldsymbol{v} + \boldsymbol{v} \otimes \boldsymbol{n}] \, \partial S \tag{40}$$

where the integral is around a closed loop, $S$, with area $A$ and normal vector $\boldsymbol{n}$, and $\boldsymbol{v}$ is the sea-ice velocity. To determine the strain rate tensor at the centers of the primary mesh, we take this integral around the edges of the cells in the primary mesh. First the cell is projected onto a flat tangent plane perpendicular to the vector joining the center of the sphere to the cell center.

We take the sea-ice velocity at a cell edge as the average of the values on the two vertices forming that edge projected onto the tangent plane:

$$\dot{\boldsymbol{\epsilon}}' = \frac{1}{A} \sum_{i}^{n_e} \frac{1}{2} [\boldsymbol{n}_i \otimes \boldsymbol{v}_i + \boldsymbol{v}_i \otimes \boldsymbol{n}_i] \, l_i \tag{41}$$





Here, $A$ is the area of the primary cell, the summation is over the $n_e$ edges of the primary cell, $\boldsymbol{n}_i$ is the normal vector to the edge $i$ that lies in the tangent plane, $\boldsymbol{v}_i$ is the edge velocity and $l_i$ is the length of edge $i$. We use the tangential projection of the velocity and account for metric terms separately. The full strain rate tensor including these metric terms is (Batchelor, 1967):

$$\dot{\epsilon}_{11} = \dot{\epsilon}'_{11} - \frac{v \tan\theta}{r}$$

$$\dot{\epsilon}_{22} = \dot{\epsilon}'_{22}$$

$$\dot{\epsilon}_{12} = \dot{\epsilon}'_{12} + \frac{u \tan\theta}{2r} \tag{42}$$

where the prime symbol signifies a strain rate without metric terms. The stress, which is determined from the strain rate tensor using the constitutive relation, is now defined on cell centers. To find its divergence we use the divergence theorem:

$$\iint \boldsymbol{\nabla} \cdot \boldsymbol{\sigma} \, \mathrm{d}A = \oint [\boldsymbol{\sigma} \cdot \boldsymbol{n}] \, \mathrm{d}l \tag{43}$$

or

$$\boldsymbol{\nabla} \cdot \boldsymbol{\sigma} = \lim_{A \to 0} \frac{1}{A} \oint [\boldsymbol{\sigma} \cdot \boldsymbol{n}] \, \mathrm{d}l \tag{44}$$

for the stress divergence at a point. The divergence of internal stress is determined at primary cell vertices (where the velocity is defined and momentum equation solved) by performing a sum around the edges of the dual mesh on a tangent projected plane, tangential to the primary cell vertex. The vertices of the dual mesh are the cell centers of the primary mesh where the strain rate has been determined. The stress divergence at primary cell vertices is then given by

$$(\boldsymbol{\nabla} \cdot \boldsymbol{\sigma})' = \frac{1}{A_d} \sum_i^{n_c} [\boldsymbol{\sigma}_i \cdot \boldsymbol{n}_i] \, l_i \tag{45}$$

where $A_d$ is the area of the dual mesh cell, the sum is over the $n_c$ vertices of the dual mesh, $l_i$ is the length of the $i$ edge of the dual mesh, and $n_i$ is a normal vector to the $i$ edge on the projected plane. As before, this gives a result without taking into account metric effects of the mesh. With those effects (Malvern, 1969) the stress divergence is:

$$(\nabla \cdot \sigma)_u = (\nabla \cdot \sigma)'_u - \frac{2\sigma_{12} \tan\theta}{r}$$

$$(\nabla \cdot \sigma)_v = (\nabla \cdot \sigma)'_v + \frac{(\sigma_{11} - \sigma_{22}) \tan\theta}{r} \tag{46}$$

where the components of $\boldsymbol{\sigma}$ are approximated as the average of the values on the dual mesh vertices.

## 2.3 Transport

To transport sea-ice fractional area and various tracers, MPAS-Seaice uses an incremental remapping (IR) algorithm similar to that described by Dukowicz and Baumgardner (2000), Lipscomb and Hunke (2004) and Lipscomb and Ringler (2005). The Lipscomb and Hunke (2004) scheme was designed for structured quadrilateral meshes and is implemented in CICE (Hunke et al., 2015). The Lipscomb and Ringler (2005) scheme was implemented on a structured SCVT global mesh consisting of quasi-regular hexagons and 12 pentagons.





For MPAS-Seaice the IR scheme was generalized to work on either the standard MPAS mesh (hexagons and other n-gons of varying sizes, with a vertex degree of 3 as in Lipscomb and Ringler (2005)) or a quadrilateral mesh (with a vertex degree of 4 as in Lipscomb and Hunke (2004) and CICE). Since the CICE IR transport code assumes a structured mesh but MPAS meshes are unstructured, the IR scheme had to be rewritten from scratch. Most of the MPAS-Seaice IR code is now mesh-agnostic, but a small amount of code is specific to quad meshes as noted below.

Here we review the conceptual framework of incremental remapping as in Hunke et al. (2015) and describe features specific to the MPAS-Seaice implementation. IR is designed to solve equations of the form

$$\frac{\partial m}{\partial t} = -\nabla \cdot (\mathbf{u}m) \tag{47}$$

$$\frac{\partial (mT_1)}{\partial t} = -\nabla \cdot (\mathbf{u}mT_1), \tag{48}$$

$$\frac{\partial (mT_1T_2)}{\partial t} = -\nabla \cdot (\mathbf{u}mT_1T_2), \tag{49}$$

$$\frac{\partial (mT_1T_2T_3)}{\partial t} = -\nabla \cdot (\mathbf{u}mT_1T_2T_3), \tag{50}$$

where $\mathbf{u} = (x, y)$ is the horizontal velocity, $m$ is mass or a mass-like field (such as density or fractional sea-ice concentration), and $T_1$, $T_2$ and $T_3$ are tracers. These equations describe conservation of quantities such as mass and internal energy under horizontal transport. Sources and sinks of mass and tracers (e.g., ice growth and melting) are treated separately from transport.

In MPAS-Seaice, the fractional ice area in each thickness category is a mass-like field whose transport is described by Eq. (47). (Henceforth, "area" refers to fractional ice area unless stated otherwise.) Ice and snow thickness, among other fields, are type 1 tracers obeying equations of the form of Eq. (48), and the ice and snow enthalpy in each vertical layer are type 2 tracers obeying equations like Eq. (49), with ice or snow thickness as their parent tracer. When run with advanced options (e.g., active melt ponds and biogeochemistry (BGC)), MPAS-Seaice advects tracers up to type 3. Thus, the mass-like field is the "parent field" for type 1 tracers; type 1 tracers are parents of type 2; and type 2 tracers are parents of type 3.

Incremental remapping has several desirable properties for sea-ice modeling:

- It is conservative to within machine roundoff.

- It preserves tracer monotonicity. That is, transport produces no new local extrema in fields like ice thickness or internal energy.

- The reconstructed mass and tracer fields vary linearly in x and y. This means that remapping is second-order accurate in space, except where gradients are limited locally to preserve monotonicity.





- There are economies of scale. Transporting a single field is fairly expensive, but additional tracers have a low marginal
cost, especially when all tracers are transported with a single velocity field as in CICE and MPAS-Seaice.

The time step is limited by the requirement that trajectories projected backward from vertices are confined to the cells sharing the vertex (i.e., 3 cells for the standard MPAS mesh and 4 for the quad mesh). This is what is meant by incremental as opposed to general remapping. This requirement leads to a CFL-like condition,

$$\frac{\max(|\mathbf{u}|\Delta t)}{\Delta x} \leq 1, \tag{51}$$

where $\Delta x$ is the grid spacing and $\Delta t$ is the time step. For highly divergent velocity fields, the maximum time step may have to be reduced by a factor of 2 to ensure that trajectories do not cross.

The IR algorithm consists of the following steps:

1. Given mean values of the ice area and tracer fields in each grid cell and thickness category, construct linear approximations of these fields. Limit the field gradients to preserve monotonicity.

2. Given ice velocities at grid cell vertices, identify departure regions for the transport across each cell edge. Divide these departure regions into triangles and compute the coordinates of the triangle vertices.

3. Integrate the area and tracer fields over the departure triangles to obtain the area, volume, and other conserved quantities transported across each cell edge.

4. Given these transports, update the area and tracers.

Since all fields are transported by the same velocity field, the second step is done only once per time step. The other steps are repeated for each field.

With advanced physics and BGC options, MPAS-Seaice can be configured to include up to ∼40 tracer fields, each of which is advected in every thickness category, and many of which are defined in each vertical ice or snow layer. In order to accommodate different tracer combinations and make it easy to add new tracers, the tracer fields are organized in a linked list that depends on
which physics and BGC packages are active. The list is arranged with fractional ice area first, followed by the type 1 tracers, type 2 tracers, and finally type 3 tracers. In this way, values computed for parent tracers are always available when needed for computations involving child tracers.

The MPAS-Seaice version of the incremental remapping transport scheme has several advantages over the one in CICE. First, as already mentioned, the scheme has been generalized to work for SCVT as well as quadrilateral meshes. Second, the
335 MPAS-Seaice scheme treats $T_3$ tracers more accurately by using more accurate integration formulas (see Eq. (60)).

We next describe the IR algorithm in detail, pointing out features that are new in MPAS-Seaice.

### 2.3.1 Reconstructing area and tracer fields

The fractional ice area and all tracers are reconstructed in each grid cell (quadrilaterals, hexagons or other $n$-gons) as functions of $\mathbf{r} = (x, y)$ in a cell-based coordinate system. On spherical grids, $\mathbf{r}$ lies in a local plane that is tangent to the sphere at the





Voronoi cell center. The state variable for ice area, denoted as $\bar{a}$, should be recovered as the mean value when integrated over the cell:

$$\int_A a(x,y)dA = \bar{a}A_C, \tag{52}$$

where $A_C$ is the grid cell area. Equation (52) is satisfied if $a(\mathbf{r})$ has the form

$$a(\mathbf{r}) = \bar{a} + \alpha_a \nabla a \cdot (\mathbf{r} - \bar{\mathbf{r}}), \tag{53}$$

where $\nabla a$ is a cell-centered gradient, $\alpha_a$ is a coefficient between 0 and 1 that enforces monotonicity, and $\bar{\mathbf{r}}$ is the cell centroid:

$$\bar{\mathbf{r}} = \frac{1}{A_C} \int_A \mathbf{r}dA. \tag{54}$$

Similarly, tracer means should be recovered when integrated over a cell:

$$\int_A a(\mathbf{r})T_1(\mathbf{r})dA = \bar{a}\bar{T}_1 A_C,$$

$$\int_A a(\mathbf{r})T_1(\mathbf{r})T_2(\mathbf{r})dA = \bar{a}\bar{T}_1\bar{T}_2 A_C,$$

$$\int_A a(\mathbf{r})T_1(\mathbf{r})T_2(\mathbf{r})T_3(\mathbf{r})dA = \bar{a}\bar{T}_1\bar{T}_2\bar{T}_3 A_C. \tag{55}$$

These equations are satisfied when the tracers are reconstructed as

$$T_1(\mathbf{r}) = \bar{T}_1 + \alpha_{T1}\nabla T_1 \cdot (\mathbf{r} - \tilde{\mathbf{r}}_1),$$

$$T_2(\mathbf{r}) = \bar{T}_2 + \alpha_{T2}\nabla T_2 \cdot (\mathbf{r} - \tilde{\mathbf{r}}_2),$$

$$T_3(\mathbf{r}) = \bar{T}_3 + \alpha_{T3}\nabla T_3 \cdot (\mathbf{r} - \tilde{\mathbf{r}}_3), \tag{56}$$

where the tracer barycenter coordinates $\tilde{\mathbf{r}}_n$ are given by

$$\tilde{\mathbf{r}}_1 = \frac{1}{\bar{a}A_C} \int_A \mathbf{r}adA,$$

$$\tilde{\mathbf{r}}_2 = \frac{1}{\bar{a}\bar{T}_1 A_C} \int_A \mathbf{r}aT_1dA,$$

$$\tilde{\mathbf{r}}_3 = \frac{1}{\bar{a}\bar{T}_1\bar{T}_2 A_C} \int_A \mathbf{r}aT_1T_2dA. \tag{57}$$

The integrals in Eq. (57) can be evaluated by applying quadrature rules for linear, quadratic and cubic polynomials as described in Section 2.3.3.

Monotonicity is enforced by van Leer limiting (van Leer, 1979). The reconstructed area and tracers are evaluated at cell vertices, and the coefficients $\alpha$ are reduced as needed so that the reconstructed values lie within the range of the mean values in the cell and its neighbors. When $\alpha = 1$, the reconstruction is second-order accurate in space. When $\alpha = 0$, the reconstruction reduces locally to first-order.



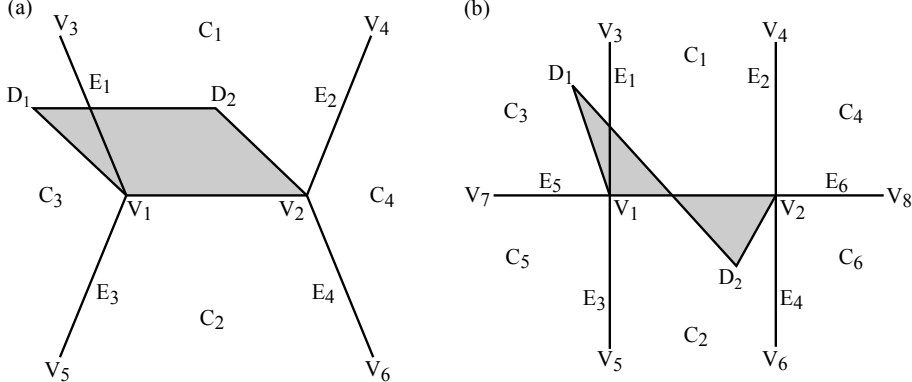

**Figure 4.** (a): Schematic showing transport across a cell edge on a standard MPAS mesh with 3 edges meeting at each vertex. The letters $C$, $E$ and $V$ denote cell centers, edges and vertices, respectively. Points $D_1$ and $D_2$ are backward trajectories, and the departure region is shaded. (b): Schematic showing transport across a cell edge on a quadrilateral MPAS mesh with 4 edges meeting at each vertex. Shaded regions are the departure regions.

### 2.3.2 Locating departure triangles

The next step is to identify the departure region associated with fluxes across each cell edge, and to divide the departure region into triangles. Figure 4a illustrates the geometry for the standard MPAS mesh. The edge has vertices $V1$ and $V2$. Each edge is oriented such that one adjacent cell ($C_1$) is defined to lie in the left half-plane and the other ($C_2$) in the right half-plane. The departure points $D_1$ and $D_2$ are found by projecting velocities backward from $V_1$ and $V_2$. The shaded departure region is a quadrilateral containing all the ice transported across the edge in one time step. In addition to $C_1$ and $C_2$, the departure region can include side cells $C_3$ and $C_4$. The side cells share edges $E_1$ to $E_4$ and vertices $V_3$ to $V_6$ with the central cells $C_1$ and $C_2$.

The edges and vertices in Fig. 4a are defined in a coordinate system lying in the local tangent plane at the midpoint of the main edge, halfway between $V_1$ and $V_2$. These coordinates are pre-computed at initialization. During each time step, departure triangles are found by locating $D_1$ and $D_2$ in this coordinate system, and then looping through the edges to identify any intersections of line segment $\overline{D_{12}}$ (i.e., the segment joining $D_1$ and $D_2$) with the various edges. If $\overline{D_{12}}$ intersects the main edge, then the departure region consists of two triangles (one each in $C_1$ and $C_2$) rather than a quadrilateral (as shown in Fig. 4b). If $\overline{D_{12}}$ intersects any of edges $E_1$ to $E_4$, the departure region includes triangles in side cells.

Each departure triangle lies in a single grid cell, and there are at most four such triangles. There are two triangles in the central cells (either one each in $C_1$ and $C_2$, or a quadrilateral that can be split into two triangles), and up to two triangles in side cells. The triangle vertices are a combination of cell vertices ($V_1$ and $V_2$), departure points ($D_1$ and $D_2$), and intersection points (points where $\overline{D_{12}}$ crosses an edge).

Figure 4b shows the geometry for a quadrilateral mesh. In this figure the departure region consists of two triangles, but it could also be a quadrilateral as in Fig. 4a. For the quad mesh there are two additional side cells ($C_5$ and $C_6$), edges ($E_5$ and $E_6$) and vertices ($V_7$ and $V_8$). The search algorithm is designed such that the code used to find departure triangles for the standard





mesh is also applied to the quad mesh. For quad meshes only, there is additional logic to find intersection points and triangles associated with the extra edges and cells. This is the only mesh-specific code in the run-time IR code. For the quad mesh there are at most six departure triangles: two in the central cells and one in each of the four side cells. If the edges meet at right angles as shown in the figure, the maximum is five triangles, but this is not a mesh requirement.

Once triangle vertices have been found in edge-based coordinates, they are transformed to cell-based coordinates, i.e., coordinates in the local tangent plane of the cell containing each triangle. (Coefficients for these transformations are computed at initialization.) Triangle areas are computed as

$$A_T = \frac{1}{2} |(x_2 - x_1)(y_3 - y_1) - (y_2 - y_1)(x_3 - x_1)| \tag{58}$$

where the $(x_i, y_i)$ are the triangle vertices.

### 2.3.3 Integrating the transport

Next, ice area and area-tracer products are integrated in each triangle. The integrals have the form of Eq. (52) for area and Eq. (55) for tracers. Since each field is a linear function of $(x, y)$ as in Eq. (53) and Eq. (56), the area-tracer products are quadratic, cubic and quartic polynomials, respectively, for tracers of type 1, 2 and 3.

The integrals can be evaluated exactly by summing over values at quadrature points in each triangle. Polynomials of quadratic or lower order are integrated using the formula

$$I = \frac{A_T}{3} \sum_{i=1}^{3} f(\mathbf{x}'_i). \tag{59}$$

The quadrature points are located at $\mathbf{x}'_i = (\mathbf{x_0} + \mathbf{x}_i)/2$, where $\mathbf{x_0}$ is the triangle midpoint and $\mathbf{x_i}$ are the three vertices. The products involving type-2 and type-3 tracers are cubic and quadratic polynomials, which can be evaluated using a similar formula with 6 quadrature points:

$$I = A_T \left[ w_1 \sum_{i=1}^{3} f(\mathbf{x}_{1i}) + w_2 \sum_{i=1}^{3} f(\mathbf{x}_{2i}) \right], \tag{60}$$

where $\mathbf{x_{1i}}$ and $\mathbf{x_{2i}}$ are two sets of three quadrature points, arranged symmetrically on the three medians of the triangle, and $w_1$ and $w_2$ are weighting factors. Coefficients and weighting factors for these and other symmetric quadrature rules for triangles were computed by Dunavant (1985). These integrals are computed for each triangle and summed over edges to give fluxes of ice area and area-tracer products across each edge.

### 2.3.4 Updating area and tracer fields

The area transported across edge $k$ for a given cell can be denoted as $\Delta a_k$, and the area-tracer products as $\Delta(aT_1)_k$, $\Delta(aT_1T_2)_k$ and $\Delta(aT_1T_2T_3)_k$. The new ice area at time $n+1$ is given by

$$a^{n+1} = a^n + \frac{1}{A_C} \sum_k \pm \Delta a_k, \tag{61}$$





where the sum is taken over cell edges $k$, with a positive sign denoting transport into a cell and a negative sign denoting outward transport. The new tracers are given by

$$T_1^{n+1} = \frac{a^n T_1^n + \frac{1}{A_C} \sum_k \pm \Delta(aT_1)_k}{a^{n+1}},$$

$$T_2^{n+1} = \frac{a^n T_1^n T_2^n + \frac{1}{A_C} \sum_k \pm \Delta(aT_1 T_2)_k}{a^{n+1} T_1^{n+1}},$$

$$T_3^{n+1} = \frac{a^n T_1^n T_2^n T_3^n + \frac{1}{A_C} \sum_k \pm \Delta(aT_1 T_2 T_3)_k}{a^{n+1} T_1^{n+1} T_2^{n+1}}. \tag{62}$$

Dukowicz and Baumgardner (2000) showed that Eq. (62) satisfies tracer monotonicity, since the new-time tracer values are
415 area-weighted averages of old-time values.

### 2.4 Column physics

CICE has sophisticated vertical physics and biogeochemical schemes, which include vertical thermodynamics schemes (Bitz and Lipscomb, 1999; Turner et al., 2013; Turner and Hunke, 2015), several melt-pond parameterizations (Flocco et al., 2010; Holland et al., 2012; Hunke et al., 2013), a delta-Eddington radiation scheme (Briegleb and Light, 2007; Holland et al., 2012),
schemes for transport in thickness space (Lipscomb, 2001), representations of mechanical redistribution (Lipscomb et al., 2007), and sea-ice BGC (Jeffery and Hunke, 2014; Jeffery et al., 2016, 2020). To include these developments in MPAS-Seaice, the column physics and BGC in CICE has been extracted into a separate library, which performs column calculations on individual grid cells with no knowledge of the details of the horizontal mesh used by the host model. This column package, now called IcePack, is used by the most recent version of CICE (Hunke et al., 2018). MPAS-Seaice uses a version of the
column package that was forked from CICE version 5.1.2.

## 3 Model validation

### 3.1 Velocity solver

To validate our implementation of the MPAS-Seaice velocity solver we investigate several idealized test cases.

### 3.1.1 Operators on planar meshes

In the first test case, we define analytical test fields on a unit square planar mesh with regular shaped cells and determine the error generated by the strain and stress divergence operators when acting on these fields. Since the strain rate operators consist of a combination of spatial derivatives in the $x$ and $y$ directions, we examine the error generated by these spatial derivative operators instead of directly by the strain rate operator. We examine meshes with both square and hexagonal cells. The test analytical field for which spatial derivatives are calculated is given by

$$f(x,y) = \sin(2A\pi x)\sin(2A\pi y) \tag{63}$$





with $A = 2.56$. This field gives sufficient variation to provide an adequate test of the operators, while the symmetry of $f(x, y)$ between the $x$ and $y$ directions allows an accurate comparison between the $\frac{\partial}{\partial x}$ and $\frac{\partial}{\partial y}$ operators. Figure 5 shows the scaling of the $L_2$ error norm for the spatial derivatives against grid resolution. As for all test cases presented in this work, the $L_2$ error norm was calculated from

$$L_2 = \sqrt{\frac{\sum A_i (f_i - \hat{f}_i)^2}{\sum A_i \hat{f}_i{}^2}} \qquad (64)$$

where the sum is over either grid cells, or vertices in the region of interest, $A_{di}$ is the area of either the primary cell, or the dual cell surrounding the vertex, and $f_i$ and $\hat{f}_i$ are the model and analytical values of the field of interest, respectively. Each doubling of resolution should reduce the $L_2$ error norm by a factor of 2 for first-order accurate schemes and a factor of 4 for second-order accurate schemes (Oberkampf and Roy, 2010). Figure 5 also shows idealized first and second-order scaling gradients as dotted lines. Evident from the figure is that the spatial derivative operators for the variational methods (Wachspress and PWL) are only first-order accurate for both square and hexagonal meshes, while the weak derivative operators are second-order accurate with much lower errors than the variational methods. This is understandable since the variational derivative operators use a one sided stencil with respect to the point where the derivatives are calculated, while the weak operator stencil surrounds the derivative location. Small differences exist between the Wachspress and PWL variational basis functions as well. Differences between errors generated for the $x$ and $y$ spatial derivatives are evident for the hexagonal mesh. These differences are caused by a difference in spatial symmetry of the mesh in these two directions for the hexagonal mesh, whereas the square mesh has the same spatial symmetry in both directions. These results are confirmed by an error analysis with a Taylor series expansion of the methods. One possible way to improve the accuracy of the variational operators is to average for each surrounding cell the different one-sided values of the derivatives calculated for a single vertex to create a multi-sided stenciled operator from the one-sided variational ones. Figure 5 shows the error scaling for this averaging method for the Wachspress ("Wachs. Avg") and PWL ("PWL Avg") basis functions. Second-order convergence is achieved in the $y$ direction with this averaging, but interestingly, only averaging the PWL basis function results in second-order convergence in the $x$ direction, while the Wachspress basis function shows only first-order convergence with much larger errors in the $x$ direction. This difference between the two directions is explained by the different geometric symmetry found in the two directions, with, for this test case, the $x$ direction parallel to a line from a cell center to a vertex and the $y$ direction parallel to a line from a cell center to edge center. The derivatives at cell centers calculated by the weak method can also be averaged to the cell vertices to create another operator ("Weak Avg" in Fig. 5), creating a second-order accurate method.

Next, with the same meshes we examine the error scaling of the stress divergence operator. The same function, $f(x, y)$, as above is used for the $u$ and $v$ components of velocity, and the analytical strain rates are calculated from these velocity components. The analytical internal ice stresses and stress divergence are calculated from this strain rate assuming a linear constitutive relation of the form $\sigma_{ij} = \dot{\epsilon}_{ij}$. The analytical internal stresses are used as input to the stress divergence operators and the output is compared to the analytical stress divergence. Figure 6 shows the scaling of the $L_2$ error norm of the stress divergence operators with grid resolution. Both the variational and weak methods show second-order convergence of errors for the square cell mesh, with the weak scheme showing significantly lower errors. For the hexagonal cell mesh the variational





**Figure 5.** Scaling of $L_2$ error norm against grid resolution for the model calculation of the gradient of an analytical function in the (a,c) $x$, and (b,d) $y$ directions for a planar test case with a regular (a,b) square cell and (c,d) hexagonal cell mesh. Grid resolution is the mean distance between cell centers. Ideal first and second-order scaling gradients are also shown as dotted lines.



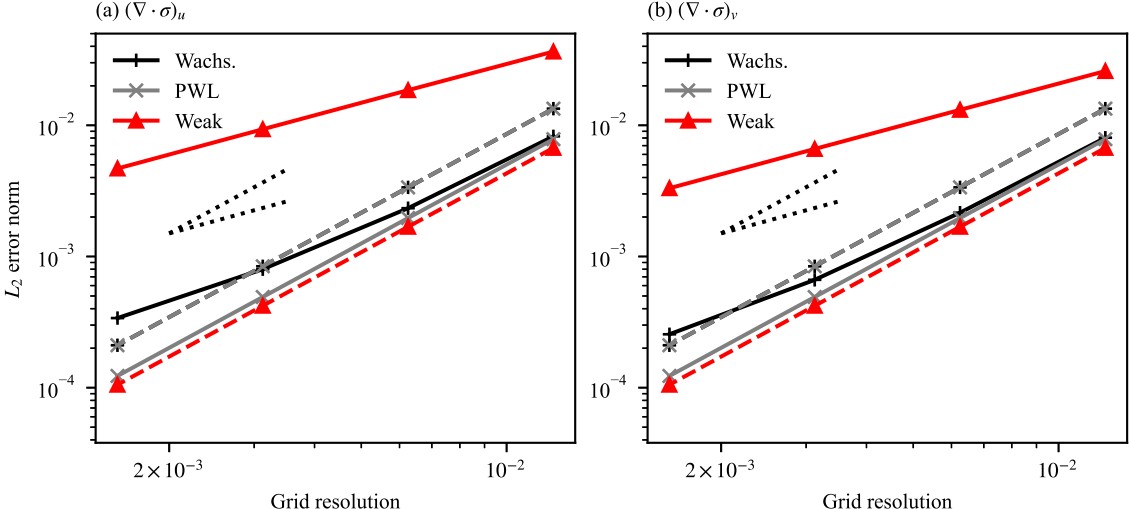

**Figure 6.** Scaling of $L_2$ error norm against grid resolution for the model calculation of the (a) $u$, and (b) $v$ components of stress divergence from an analytical strain rate tensor field for a planar test case with a regular mesh. Grid resolution is the mean distance between cell centers. Solid lines denote hexagonal cell meshes, whereas dashed lines signify square cell meshes. Ideal first and second-order scaling gradients are also shown as dotted lines.

methods show lower errors with better convergence rates than the weak method. With the PWL basis functions the variational method shows second-order convergence, while with Wachspress basis functions the variational method shows a varying order of convergence with near second-order convergence at low resolution but with the order decreasing to first with higher resolutions. This suggests there is a small source of first-order error when using the Wachspress basis functions. For the hexagonal cell mesh, the weak method shows only first-order convergence with significantly higher error than the variational method. This is because the line integral around vertices for the weak method involves an integral around a triangle. Unlike the integral around a square or hexagon, there are no opposite sides to a triangle. For the weak integrals around squares or hexagons, opposite edges of the polygon lead to cancellation of first-order error, which does not occur for integrals around triangles. Smaller differences between the $x$ and $y$ directions are visible than with the spatial derivative operators examined above. Again, these results are confirmed with a Taylor series expansion of the methods.

In summary, for regular planar meshes, for the gradient operators, the weak and averaged PWL and averaged weak schemes show a higher order of error convergence and lower absolute errors than the variational schemes and averaged Wacshpress scheme. Conversely, for the stress divergence operator, the variational schemes show lower absolute errors and better error convergence than the weak scheme. Within the variational scheme, use of the PWL basis functions displays more consistent error convergence characteristics than the Wachspress scheme.





### 3.1.2 Operators on a unit sphere

Having examined the spatial operators on planar meshes, we now examine the effect of metric terms introduced by using the operators on a sphere. To do this we assume an analytical velocity field on a unit sphere, and derive analytical strain rate and stress divergence fields from those velocity fields (again assuming a linear consitutive relation of the form $\sigma_{ij} = \dot{\epsilon}_{ij}$). We use spherical harmonic functions, $Y$, for the analytical velocity fields with $u(\theta, \phi) = Y_{l=5}^{m=3}(\frac{\pi}{2} - \theta, \phi)$ and $v(\theta, \phi) = Y_{l=4}^{m=2}(\frac{\pi}{2} - \theta, \phi)$ where $u$, $v$, the latitude, $\theta$, and the longitude, $\phi$, are on the rotated geographical mesh (see section 2.1). This choice of spherical harmonic, with different values of $m$ and $l$ for $u$ and $v$, produces a sufficiently varying velocity field to test the strain rate and stress divergence operators. Figures 7a,b show these analytical velocity fields, while Figs. 7c,d and Figs. 7e,f,g show the derived analytical fields for stress divergence and strain rate respectively.

Errors generated for the strain rate operators for this test case are displayed in Figs. 7h,l,p. These show that the error is significantly lower for the weak scheme compared to the two variational schemes. They also show that the error in the variational scheme consists of alternating signed error values within a cell, while the error for the weak scheme is enhanced around the twelve pentagonal cells found in the quasi-uniform SCVT mesh. These features are clearer in Figs. 7i,m,q which shows the detail around one of the pentagonal cells. A histogram of error values for the various strain rate operators is presented in Fig. 8a. Evident are the much larger error values for the variational schemes compared to the weak scheme. The scaling of $L_2$ error norm with grid resolution is shown in Fig. 9a. $L_2$ error norm is calculated for regions of the grid with latitude, $|\theta| > 20°$ (so that the poles of the rotated mesh are excluded) for four different grid resolutions. Evident from the scaling figure is that the variational schemes display first-order convergence and the weak scheme shows second-order convergence and much lower error levels. Also shown on this figure are results for the operator averaging methods described in section 3.1.1. Here the PWL and Weak operator averaging show similar error scaling characteristics with second-order convergence at low resolution, reducing to first-order convergence at higher resolution. Also apparent is that a much lower improvement in error for the Wachspress variational scheme is achieved with averaging. Figure 9c shows the scaling of the $L_\infty$ norm, defined as $\max_i |f_i - \hat{f}_i|$, which shows first-order convergence for all strain rate operator methods.

Next we examine the stress divergence operators on the unit sphere. Figures 7j,n,r show the error for the two variational schemes and the weak scheme. Away from the pentagonal cells, the variational methods display smaller errors than the weak scheme, but show a significant enhancement of error at the pentagonal cells, something not found with the weak scheme. As for the strain rate operator, the two variational stress divergence schemes show very similar error patterns on the unit sphere. These results are more easily seen in Figs. 7k,o,s which again show the detail around one of the pentagonal cells. The error enhancement around the pentagonal cell is almost entirely due to the choice of area denominator in Eq. (8). For the area denominator Fig. 7 uses the triangle area of the dual cell. Figure 10 again shows the error for the stress divergence operator on the unit sphere for the two variational schemes, but using the alternate formulation for the area denominator defined in Eq. (34). As can be seen with this alternate formulation there is no enhancement in error around the pentagonal cells. Figure 8b shows a histogram of the errors generated by the stress divergence operators. The enhanced error around the pentagonal cells for the variational scheme with the original area denominator formulation is present here as the high error tails in the distribution.





**Figure 7.** Orthographic projection of one hemisphere of the unit sphere test case described in section 3.1.2. (a) and (b) show the analytical forms of velocity used in the numerical schemes, and used to calculate the desired analytical strain rates (shown in (e-f)), and the desired analytical stress divergence (shown in (c) and (d)). (h,i,l,m,p,q) show the error in calculation of the $\dot{\epsilon}_{11}$ component of the strain rate for the (h,i) Wachspress variational scheme, the (l,m) PWL variational scheme, and the (p,q) weak scheme. (j,k,n,o,r,s) show the error in calculation of the $u$ component of the stress divergence for the (j,k) Wachspress variational scheme, the (n,o) PWL variational scheme, and the (r,s) weak scheme. Detail is shown around one of the pentagonal mesh cells for each scheme in (i,k,m,o,q,s). The error is the difference between the model and analytical values.



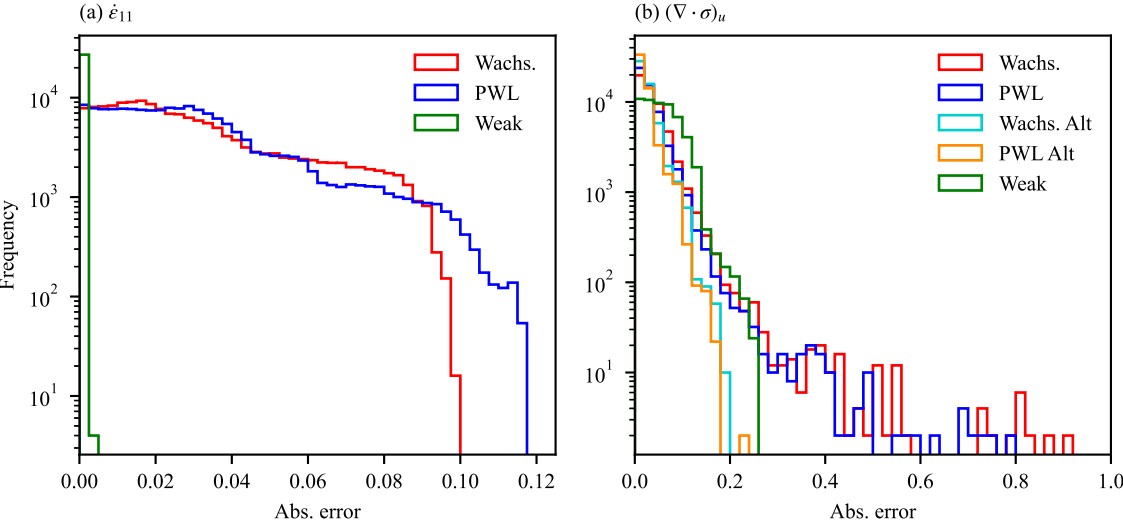

**Figure 8.** Frequency of grid cell error in the calculation of strain rate (a) and stress divergence (b) from an analytical velocity field for a spherical test case. The error is given as the absolute value of the difference between model and analytical fields. The results are plotted as histograms for the two variational schemes (using Wachspress and PWL basis functions, and the alternative area formulation) and the weak scheme.

The improvement for the alternate area denominator formulation is also apparent, with no such tails in the distribution for this formulation. Scaling for the stress divergence $L_2$ error norm against grid resolution is shown in Fig. 9b, where all methods display first-order accuracy, and the improvement in error from the alternate area denominator formulation is evident. The $L_\infty$ error norm scaling for the stress divergence operators is shown in Fig. 9d where the effect of the enhanced error for the variational schemes around pentagonal cells is evident as the failure in $L_\infty$ error convergence. The alternate area denominator methods show much better convergence at low resolutions, but convergence slows for higher resolutions. The weak method

shows linear convergence for all grid resolutions.

    Overall, the results for the unit sphere test case show similar error characteristics to the planar test case of section 3.1.1. The largest effect of moving from a regular planar mesh to a unit sphere is that the variational scheme shows significantly enhanced errors for the stress divergence operator for the irregular cells surrounding the 12 pentagonal cells present on the unit sphere. This effect is ameliorated by using the alternate area denominator scheme (Eq. (34)).

**3.1.3   Velocity solver in a square domain**

Since the MPAS framework and MPAS-Seaice support quadrilateral grids, direct comparisons can be made between MPAS-Seaice and CICE. For idealized planar test cases it is possible to set up MPAS-Seaice to be have a virtually identical velocity solver algorithm to CICE. This is achieved by using the variational scheme with Wachspress basis functions and defining the $u$ and $v$ velocity component directions in the same sense as CICE. To compare MPAS-Seaice to CICE, we use a simple test

**Figure 9.** Scaling of $L_2$ (a,b) and $L_\infty$ (c,d) error norm against grid resolution for the model calculation of (a,c) the $\dot{\epsilon}_{11}$ component of the strain rate tensor from an analytical velocity field, and (b,d) the $u$ component of the stress divergence from an analytical strain rate tensor field on the unit sphere for a spherical test case. Grid resolution is the mean distance between cell centers. Only mesh cells where $\theta > 20°$ are included. Ideal first and second-order scaling gradients are also shown.





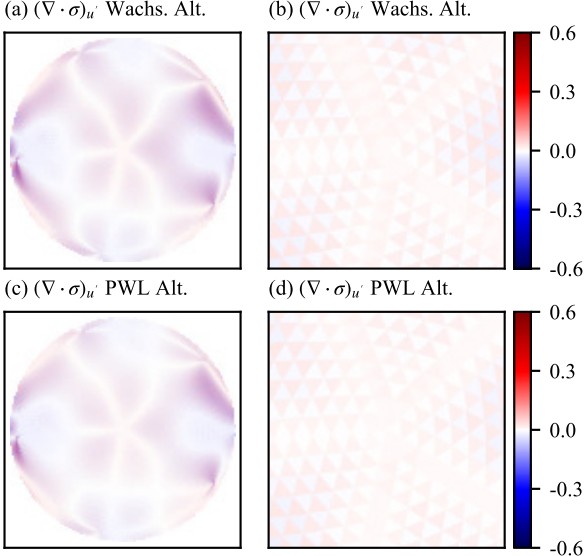

**Figure 10.** Orthographic projection of one hemisphere of the stress divergence operator error for the unit sphere test case. Shown are results for the variational schemes using the alternate area denominator formulation. (a) and (b) show results for the Wachspress variational scheme, while (c) and (d) show results for the PWL variational scheme. Detail is shown around one of the pentagonal mesh cells for each scheme in (b,d).

case, similar to that used in Hunke (2001). This test case has a square planar domain of size 80km. Ice thickness is fixed at 2m, while ice concentration increases linearly in the eastwards direction from zero at the western boundary to one at the eastern boundary, and no snow is present. Only the velocity solver is active, with no advection or column physics. The sea ice is forced by atmospheric winds and ocean currents. The atmospheric wind forcing has the form

$$u_a = 5 - 3 \sin \frac{2\pi x}{L_x} \sin \frac{\pi y}{L_y} \quad \mathrm{m\,s}^{-1} \tag{65}$$


$$v_a = 5 - 3 \sin \frac{2\pi y}{L_y} \sin \frac{\pi x}{L_x} \quad \mathrm{m\,s}^{-1} \tag{66}$$

while the ocean currents have the form

$$u_o = 0.1 \frac{2y - L_y}{L_y} \quad \mathrm{m\,s}^{-1} \tag{67}$$

$$v_o = -0.1 \frac{2x - L_x}{L_x} \quad \mathrm{m\,s}^{-1} \tag{68}$$

where $x$ and $y$ are the horizontal position and $L_x$ and $L_y$ are the domain size in the $u$ and $v$ directions respectively. These forcing velocity fields are shown in Fig. 11. Sea-ice velocity was simulated for four time steps (each of length 1 hour), which



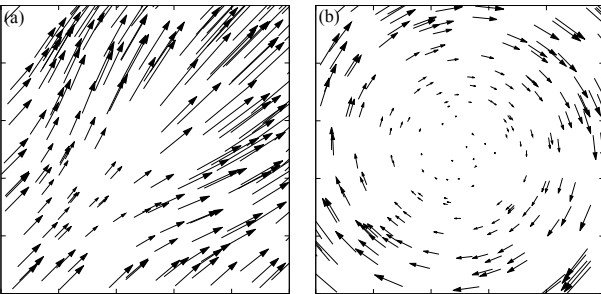

**Figure 11.** Forcing fields of (a) wind velocity and (b) ocean velocity used in a square domain test case.

was sufficient time for the ice state to relax to the elliptical yield curve. Figure 12 shows a comparison of the modeled eastwards velocity and stress divergence component between CICE and MPAS-Seaice. In this comparison MPAS-Seaice uses an identical

quadrilateral mesh to CICE. The eastwards component of wind stress pushes the sea ice against the east model boundary, and it is here that significant internal sea-ice stresses are present (see Fig. 12e). The figure also shows that the three MPAS-Seaice schemes for calculating stress divergence are capable of reproducing the results of CICE. As expected, the variational scheme with Wachspress basis function best reproduces the results of CICE, since this algorithm is most similar to CICE. Differences with CICE appear as noise, a function of incompletely damped elastic waves from the EVP rheology (Hunke, 2001). Figure 13

shows similar results for the same test case but with MPAS-Seaice using a regular hexagonal mesh. Here differences between the weak and variational scheme with the PWL basis and the variational scheme with the Wachspress basis are larger than for quadrilateral meshes, but still small. The majority of the differences between the methods, such as the blue linear feature in Figs. 13b,c, is caused by differences in calculated strain rate. Compared to the quadrilateral case, for hexagonal cells there are larger differences between the derivatives of the Wachspress and PWL basis functions at cell vertices (used in Eqs. (30)). As

can be seen from Fig. 14, all the schemes have stress states that lie within or on the elliptical yield curve for both quadrilateral and hexagonal meshes. A banding structure to the principal stresses can be seen for the quadrilateral meshes. Each band corresponds to grid cells in a vertical column in the top right-hand corner of the domain.

### 3.2 Transport

To verify that the incremental remapping transport scheme works as expected, we ran two test cases on a global spherical grid,

following Lipscomb and Ringler (2005). In each case there is a steady eastward velocity field given by $\mathbf{u} = (u_0 \cos\theta, 0)$, where $u_0 = (2\pi R)/(12 \text{ days})$ and $R$ is the Earth's radius. We first advect a circular region of ice that has initial concentration given by a cosine bell within a distance $R/3$ of a central point on the equator, and $a = 0$ elsewhere. The initial ice thickness is $h = 1$. We compare results from the IR scheme to a simple first-order upwind scheme to demonstrate the improvements in numerical diffusion gained by increasing the order of transport scheme. For both the IR scheme and the first-order upwind scheme, the

model was run at several grid resolutions for 12 days, at which time a perfect advection scheme would give a solution equal to the initial condition. For a grid resolution of 120 km, Fig. 15a shows equatorial cross sections of $a$ for the initial condition,

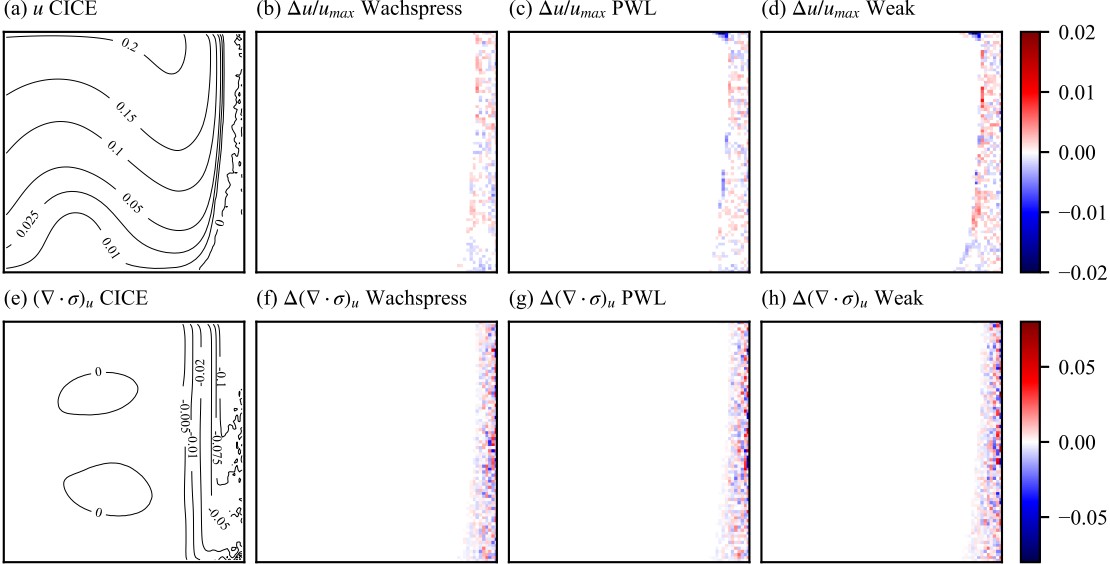

**Figure 12.** (a): $u$ velocity component of CICE for the square domain test case using a quadrilateral mesh. (b,c,d): Difference ($\Delta u/u_{max}$) between the $u$ component of MPAS-Seaice and CICE in the square domain test case for the MPAS-Seaice Wachspress variational, PWL variational and weak schemes respectively and using a quadrilateral mesh. (e-h): As (a-d) but for the $u$ component of the divergence of internal ice stress.

the upwind solution and the IR solution. Figure 16a-c shows the spatial distribution of ice concentration before and after the experiment for the same resolution. As expected, the upwind solution is very diffuse, while the IR scheme does a much better job of preserving the initial shape.

Next, we advect a slotted cylinder with initial concentration $a = 1$, initial thickness $h = 1$ and radius $R/2$, also centered on the equator. We set $a = 0, h = 0$ for $r > R/2$ and also in a slot of width $R/6$ and length $5R/6$, with the long axis perpendicular to the flow. The model was run for 12 days at several resolutions. Figure 15b shows the initial condition and the upwind and IR solutions along the equator at resolution 120 km, while Fig. 16d-f shows the spatial distribution of the ice concentration before and after the experiment for the same resolution. Again, the upwind scheme is very diffusive; all traces of the slot vanish. The

IR scheme does well at maintaining the initial plateaus and a distinct slot, although diffusion into the slot raises the minimum concentration from 0 to ~0.2. Early in the IR simulation there is truncation at the leading and trailing edges of the cylinder, where the gradient is limited, but advection continues thereafter with little change in shape. The maximum concentration is just above 1 because the discretized velocity field is slightly convergent, and diffusion is small. On a plane (not shown) with steady $\mathbf{u} = (u_0, 0)$, the discretized velocity field is non-convergent, and $a$ remains bounded by [0,1]. Ice thickness, having

been initialized to h = 1 everywhere, remains h = 1 everywhere (within roundoff), showing that both schemes preserve tracer monotonicity as expected.



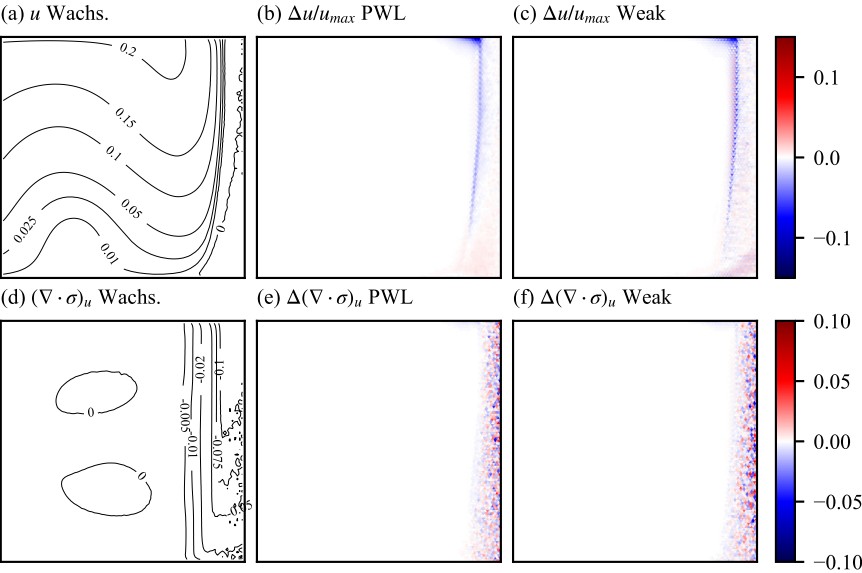

**Figure 13.** (a): $u$ velocity component of MPAS-Seaice with the Wachspress variational scheme for the square domain test case using hexagonal elements. (b,c): Difference ($\Delta u/u_{max}$) between the $u$ component of MPAS-Seaice using the Wachspress variational scheme and MPAS-Seaice using the PWL variational and weak schemes respectively in the square domain test case using hexagonal cells. (d-f): As (a-c) but for the $u$ component of the divergence of internal ice stress.

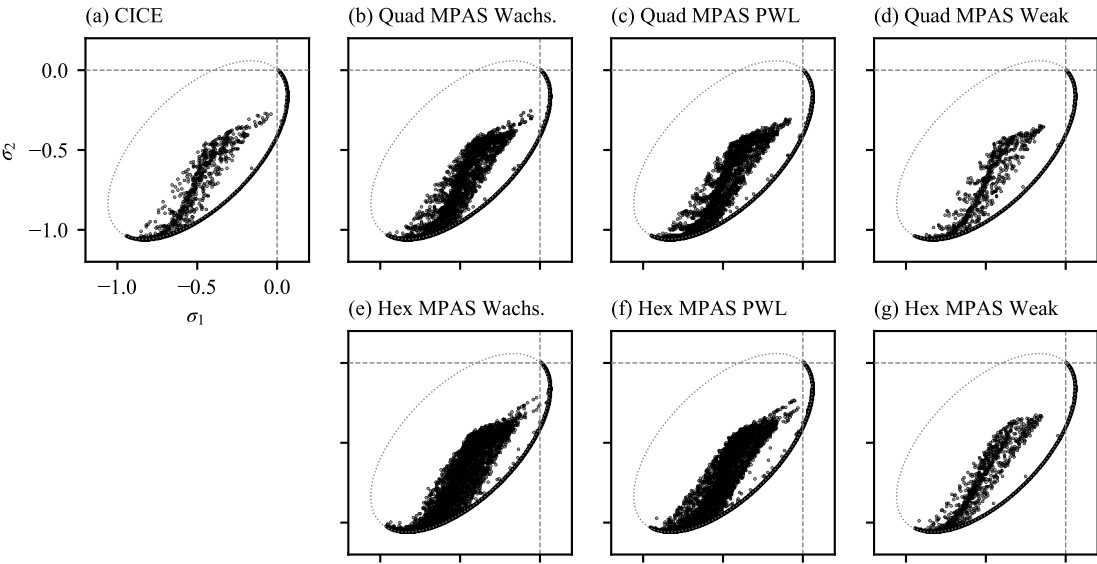

**Figure 14.** Principal stress components for the square domain test case and (dotted) the EVP yield curve. (a): CICE. (b-d): MPAS-Seaice on a quadrilateral mesh and using the Wachspress variational, PWL variational and weak schemes respectively. (e-g): MPAS-Seaice on a hexagonal mesh and using the Wachspress variational, PWL variational and weak schemes respectively.





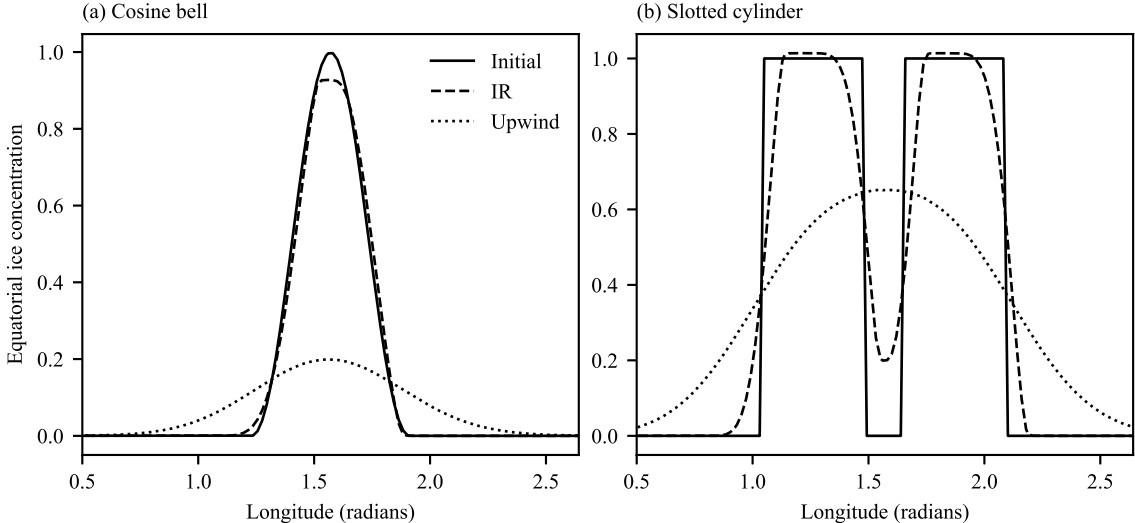

**Figure 15.** Comparison of incremental remapping to a first-order upwind scheme for advection around the sphere. (a) shows a cross section of a circular region of sea ice whose center lies on the equator, with radius $R/3$ (where $R$ is the Earth's radius) and initial sea-ice concentration given by a cosine bell. (b) shows a cross section of a slotted cylinder whose center lies on the equator, with radius $R/2$. The grid resolution is 120 km. The exact solution (which corresponds to the initial condition) is shown by a solid line, IR by long dashes, and upwind by short dashes.

Figure 17 shows the $L_2$ error norm of the 12-day solution for four grid resolutions ranging from 60 km to 480 km (where resolution is taken as the mean distance between neighboring cell centers). Figure 17 shows that the IR solution converges with close to second-order accuracy (indicated by the dotted diagonal line) for the cosine bell and converges slightly below first-order accuracy for the slotted cylinder. This slow convergence for the slotted cylinder is the result of the ice concentration discontinuity at the cylinder edge, which becomes sharper as the distance between neighboring grid cells decreases when resolution increases. The upwind scheme converges more slowly than the IR scheme, with larger errors at all resolutions. Some along-motion asymmetry is visible in the IR solutions. This is also visible in IR solutions in Lipscomb and Ringler (2005) and is expected since IR is an upstream-based method.

## 3.3 Column physics

To validate the column physics in MPAS-Seaice we make use of the fact that CICE and MPAS-Seaice share identical code. CICE and MPAS-Seaice were run with identical forcing and with dynamics disabled. Results from the two models were bit-for-bit identical, indicating a correct implementation of the column physics in MPAS-Seaice.



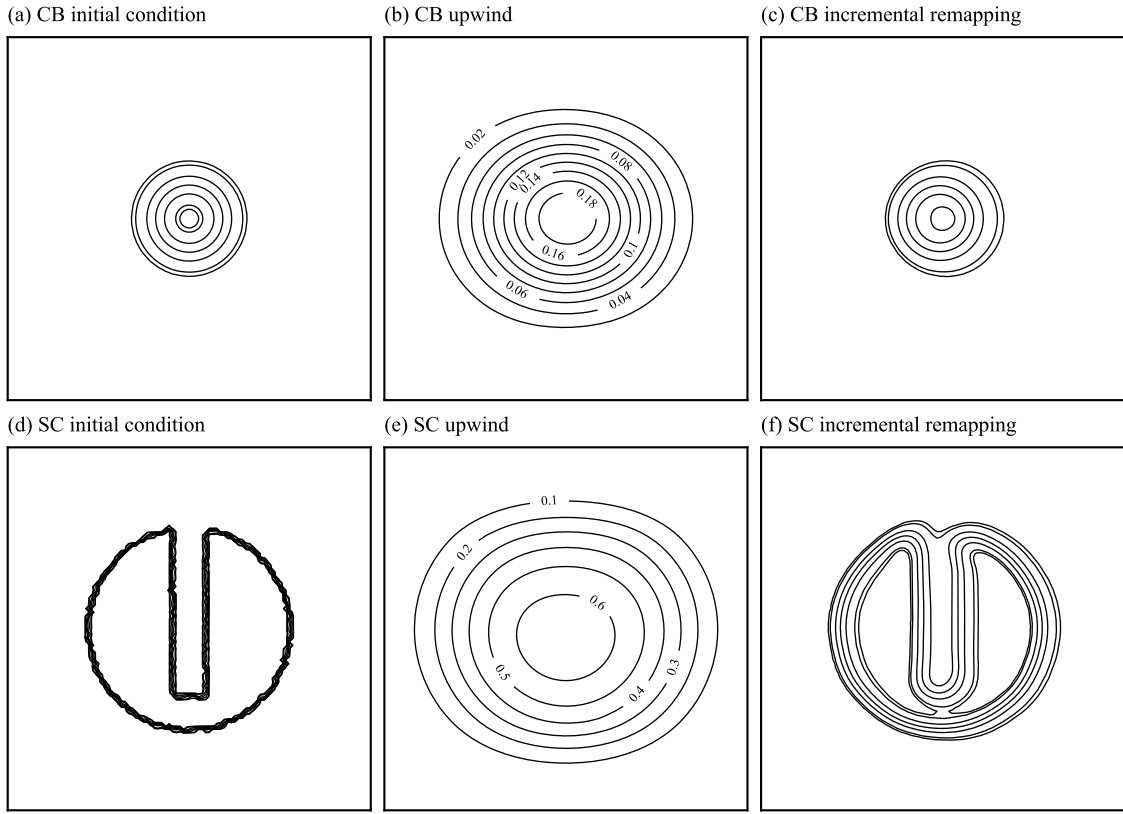

**Figure 16.** Initial (a,d) and final (b,c,e,f) ice concentration contours at120 km resolution for the cosine bell (a-c) and slotted cylinder (d-f) advection test cases. Results for the upwind advection scheme are shown in (b) and (e), while results from the incremental remapping scheme are shown in (c) and (f). Contours are at levels 0.05, 0.1, 0.3, 0.5, 0.7, 0.9, and 0.95 unless otherwise shown. The direction of transport is to the right.

## 3.4 Global simulations

To validate the full MPAS-Seaice model in a global setting we perform stand-alone global simulations, and compare simulation results to observational datasets and to the results of simulations conducted with the CICE model (Hunke et al., 2015, release version 5.1.2). We choose version 5.1.2 of CICE to compare against to keep the comparison as clean as possible, since this is the CICE version where the CICE and MPAS-Seaice column physics codes diverged. To aid the comparison to CICE we run both models with a one-degree displaced pole quadrilateral mesh. We perform the simulation for 50 years from 1958 to 2007.

Settings for the column physics are the standard ones for CICE (Hunke et al., 2015). For atmospheric and oceanic forcing we repeat the methods used by Hunke et al. (2013) and Hunke and Holland (2007). Air temperature, air specific humidity and air velocity at 10 m height and six-hourly frequency are taken from the Coordinated Ocean-ice Reference Experiments (CORE) Corrected Inter-Annual Forcing Version 2.0 (Large and Yeager, 2009; Griffies et al., 2009). Monthly climatologies of





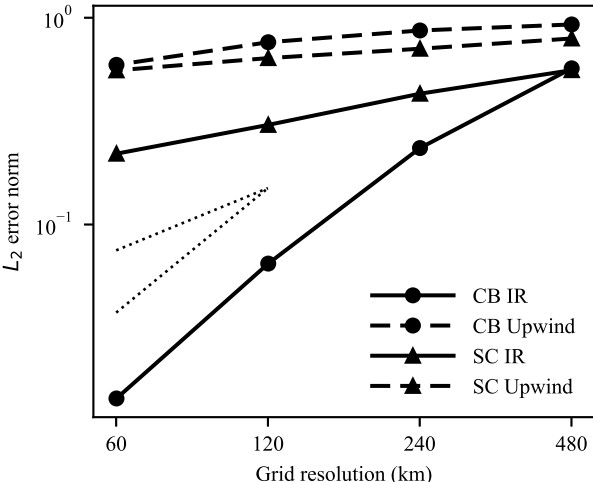

**Figure 17.** Scaling of the $L_2$ error norm with grid resolution for the cosine bell (CB) and slotted cylinder (SC) advection tests shown in Fig. 15. IR scaling is shown by the solid lines, upwind by long-dashed lines. Theoretical linear and quadratic scaling are shown by a short-dashed lines.

precipitation (Griffies et al., 2009) and cloudiness (Röske, 2001) are also used. Downwelling shortwave radiation is calculated

from the monthly climatology of cloudiness using the AOMIP shortwave forcing formula (Hunke et al., 2015). Downwelling longwave radiation is calculated according to Rosati and Miyakoda (1988). Oceanic inputs, consisting of sea surface salinity, initial sea surface temperature, currents, sea-surface slope and deep ocean heat flux, come from monthly mean output of 20 years of a Community Climate System Model (CCSM) climate run (b30.009, Collins et al., 2006). The sea surface temperature is determined by a thermodynamic ocean mixed layer parameterization as used in Hunke et al. (2013). All input forcing fields

are interpolated linearly in time, although the MPAS forcing functionality can be easily extended to allow interpolation in time with arbitrary order. To get good agreement between CICE and MPAS-Seaice it was necessary to fix several implementation errors in the CICE 5.1.2 forcing scheme. First, CICE 5.1.2 incorrectly repeats the rotation from geographical to coordinate directions for ocean current climatology data. Second, the ocean current forcing routine, rather than reading the surface layer ocean current data for all twelve months of the climatology, instead reads in the first twelve vertical layers for January. These

issues have been fixed in CICE 6+.

Figure 18a-f compares total sea-ice extent between the MPAS-Seaice and CICE models and observational values for the years 1988 to 2007 inclusive. The observational sea-ice extent values for the Northern (Cavalieri and Parkinson, 2012; Parkinson et al., 1999) and Southern Hemisphere (Parkinson and Cavalieri, 2012; Zwally et al., 2002) show excellent agreement with both models, with the seasonal cycle of sea-ice extent well represented in both hemispheres. The largest discrepancy occurs in

the Southern Hemisphere where austral summertime sea-ice extent is too low in both models.

Figure 19 shows a similar agreement, comparing sea-ice concentration from SSMI observations using the NASATeam method (Cavalieri et al., 1996, updated yearly) to model results for summer and winter periods in both hemispheres. Minor



**Figure 18.** Sea-ice extent (area with ice concentration greater than 15%) by year for the Northern (a-c) and Southern (d-f) Hemispheres, for MPAS-Seaice (a,d), CICE (b,e) and SSMI satellite observations (c,f). Northern Hemisphere total sea-ice volume by year for (g) MPAS-Seaice, (h) CICE, and (i) PIOMAS.



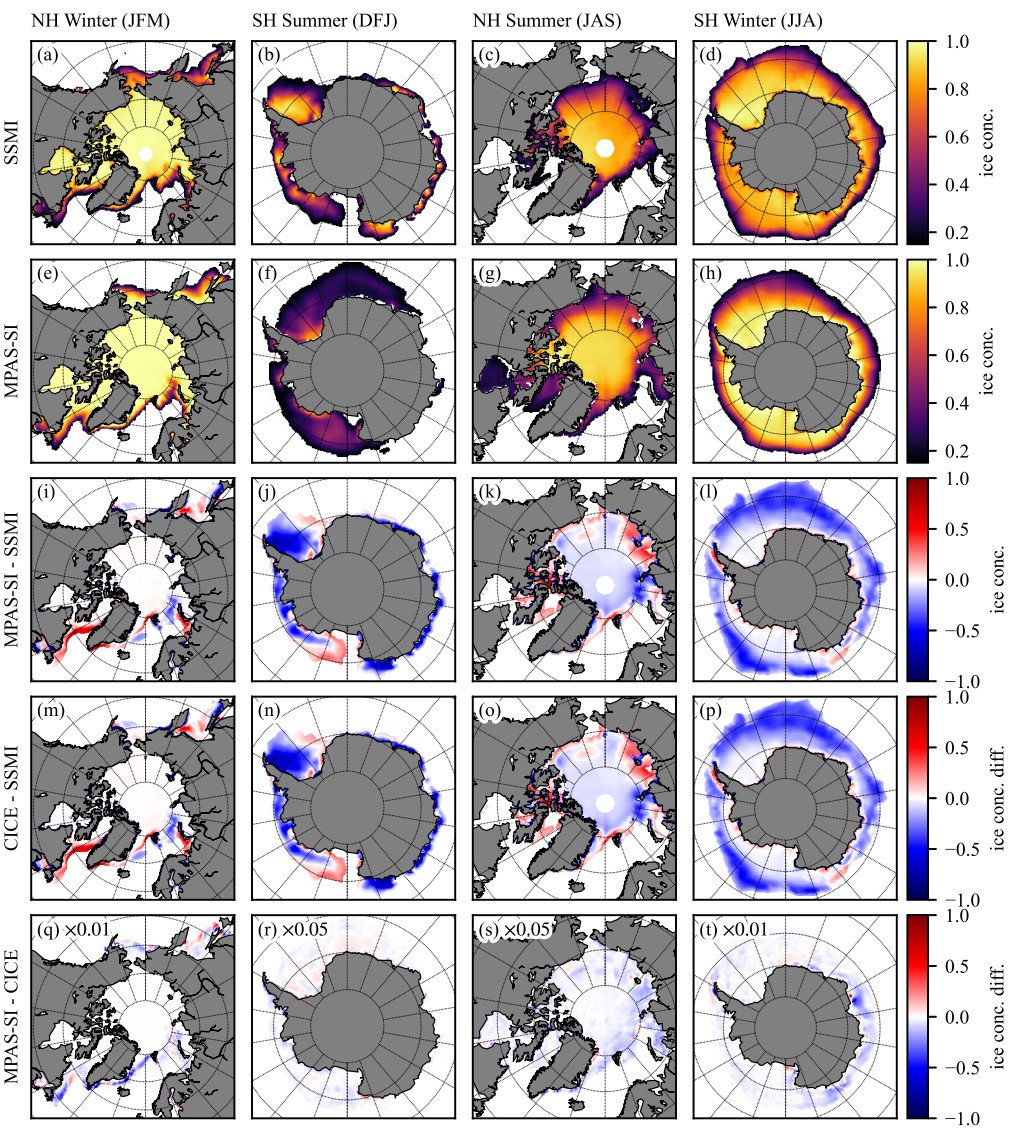

**Figure 19.** Spatial climatological maps for 1988 to 2007 of sea-ice concentration from (a-d) SSMI satellite observations processed with the NASATeam algorithm, and (e-h) MPAS-seaice. Differences between (i-l) MPAS-Seaice, and (m-p) CICE ice concentration and SSMI observations. (q-t): Differences between MPAS-Seaice and CICE ice concentration. (a,e,i,m,q): Northern Hemisphere Winter: January, February, and March. (b,f,j,n,r): Southern Hemisphere summer: December, January, and February. (c,g,k,o,s): Northern Hemisphere summer: July, August, and September. (d,h,l,p,t): Southern Hemisphere winter: June, July, and August.





differences are present in both models at the Arctic ice edge during winter and in the pack interior in summer. In general the sea-ice extent is well reproduced. In the Southern Hemisphere the sea-ice extent is reasonably reproduced in summer by both models with more significant differences in the pack interior. As expected from Fig. 18, larger differences are found between the model results and observations in the Southern Hemisphere summer where ice concentration is particularly underrepresented in the models in the Weddell Sea. In general, agreement is much closer between the two models than between the models and observations. This is expected given the similarity of the models and model forcing. Differences between MPAS-Seaice and CICE are explained by a number of differences in implementation between these models. Firstly, since MPAS-Seaice culls land cells, interpolation between T and U points (cell centers and vertices in MPAS parlance) does not include zero values for land cells, unlike in CICE. Weights in this interpolation also do not sum exactly to one for CICE, since the CICE interpolation scheme mixes T and U cell areas. Secondly, CICE determines the grid angle with respect to geographical coordinates for its T points from averaging over the angle values for surrounding U points. This generates errors in wind and ocean current forcing directions around the North Pole. Finally, for ocean forcing, MPAS-Seaice and CICE have different orders of operation for interpolation in time, space, and rotation from geographical to model coordinate directions, generating small differences in forcing values.

Total sea-ice volume for the Northern Hemisphere is compared between the models and the Pan-Arctic Ice Ocean Modeling and Assimilation System (PIOMAS) assimilated data product (Schweiger et al., 2011) in Fig. 18g-i. Both models and PIOMAS have the expected seasonal cycle with a similar variation between summer and winter and a decrease in total volume over time. Only small differences exist between the two models and the PIOMAS product in terms of absolute ice volume. Figure 20 shows the spatial patterns of sea-ice thickness for the Northern Hemisphere in summer, autumn and winter, compared to observations of sea-ice thickness from ICESat (see Fig. 20, Yi and Zwally, 2009). ICESat data is available from set periods from 2003 to 2008 during these seasons, and the model climatological maps are generated for the same periods. ICESat observations exclude sea ice with concentration less than 20%, so sea-ice thicknesses were excluded from the model results in the comparison where model ice concentration was less than 20%. Similar spatial patterns of sea ice are found in the results of both models and the ICESat observations, with thicker sea ice along the Canadian archipelago coast and thinner sea ice everywhere at the end of the summer melt season. Both MPAS-Seaice and CICE have excess sea-ice thickness compared to ICESat observations in the Beaufort sea and western Arctic basin and a deficit of sea-ice thickness in the Eurasian basin.

Finally, we perform simulations with MPAS-Seaice on a quasi-uniform SCVT mesh with 30km cell separation. The mesh is prepared using the *Jigsaw* tool (Engwirda, 2017) and the equatorial region is culled for computational efficiency. Figure 21 shows differences in Northern and Southern hemispheric total sea-ice extent and volume between this 30km SCVT and the one-degree quadrilateral mesh used above. Results are shown as a percent difference between the meshes with both simulations using the Wachspress variational scheme. Agreement is generally very good with only a few percent differences in extent and volume between the meshes, with the quadrilateral mesh having smaller extent than the SCVT mesh in the Northern Hemisphere and larger extent in the Southern Hemisphere, while volume is lower in both hemispheres for the quadrilateral mesh. The differences also have a strong seasonal variation. We compare several of the other operator methods with the Wachspress variational scheme, all using the quasi-uniform SCVT mesh. While using the alternate area denominator significantly reduced

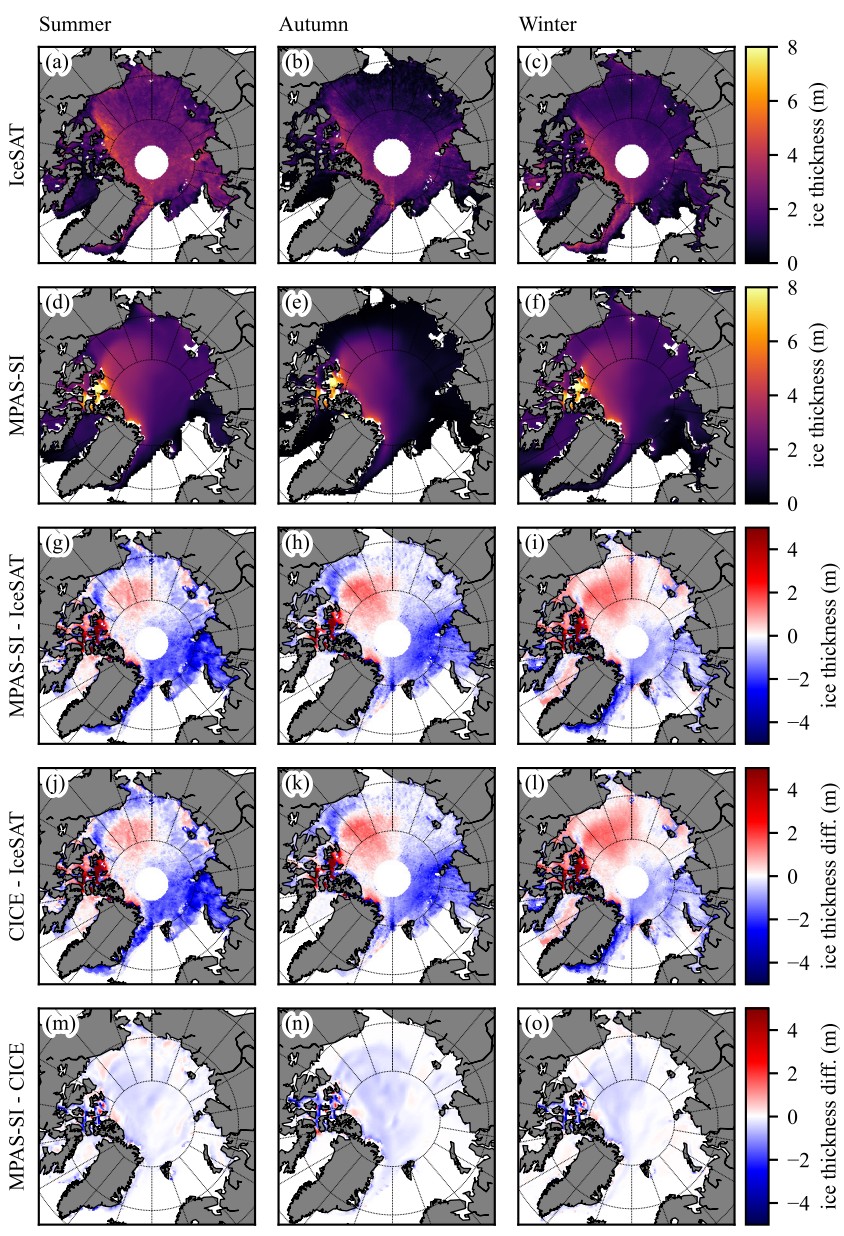

**Figure 20.** Spatial climatological maps of sea-ice thickness from ICESat (a-c), and MPAS-Seaice (d-f). Differences between (g-i) MPAS-Seaice, and (j-l) CICE ice thickness and ICESat observations. (m-o): Differences between MPAS-Seaice and CICE ice thickness. (a,d,g,j,m): NH summer. (b,e,h,k,n): NH Autumn. (c,f,i,l,o): NH Winter.





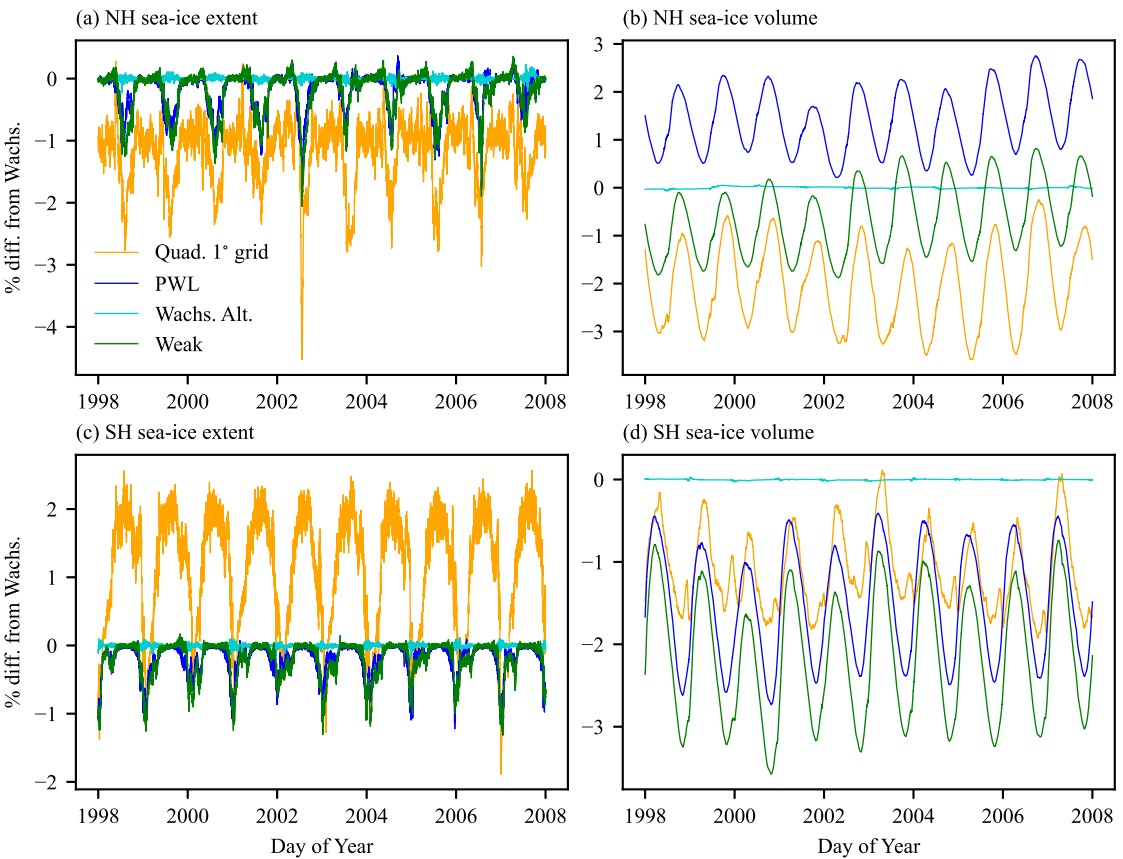

**Figure 21.** Difference (as %) in hemispheric sea-ice extent (a,c) and volume (b,d) between global simulations using the Wachspress variational scheme with a quasi-uniform 30 km SCVT mesh and several other simulations. Results are shown for the Northern (a,b) and Southern (b,d) hemispheres.

the errors surrounding pentagonal cells in the unit sphere test case in section 3.1.2, it has almost no effect on total ice extent or volume in basin-scale simulations. Compared to the Wachspress variational scheme, the PWL variational and weak schemes

have less effect on ice extent and a similar effect on ice volume as compared to the difference between the quasi-uniform SCVT and one degree quadrilateral meshes. Differences are also strongly seasonal for these changes in operator, especially for total volume.

## 4  Computational Performance

Through the MPAS framework, MPAS-Seaice incorporates code parallelization through domain decomposition and message

passing with the Message Passing Interface (MPI) library. To assess the computational performance of MPAS-Seaice when run on multiple processors, we perform a strong-scaling performance analysis (Fig. 22). Fixing the grid resolution and run duration,



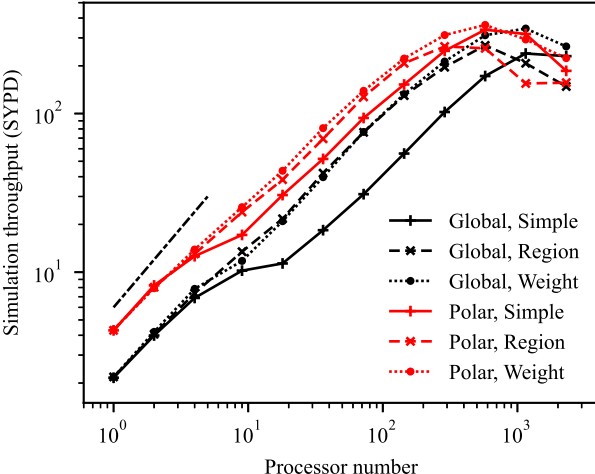

**Figure 22.** Strong-scaling performance characteristics of MPAS-Seaice for a global QU60km mesh (*Global*) and a QU60km mesh with equatorial cells culled (*Polar*). Simulated years per day of wall clock time (SYPD) is plotted against the number of compute nodes. Perfect strong-scaling would be linear. Three partitioning methods are employed as described in the main text: *Simple*, *Region*, and *Weight*.

we vary the number of processors and compare the time taken to perform the simulation. We measure model performance in simulated years per day of wall clock time (SYPD). The SYPD metric excludes time spent initializing or finalizing the model simulation, but includes time spent reading in time-varying forcing data. No output data files are written during these

simulations. Simulations were performed as in section 3.4 with a duration of 10 days. Two MPAS meshes are compared. The first is a global quasi-uniform SCVT mesh with cell-to-cell distance of 60km (QU60km). This mesh has 114,539 cell centers and 234,609 vertices. The MPAS framework allows arbitrary regions of the domain to be removed. We use this capability with MPAS-Seaice by removing equatorial cells where sea ice does not form. This significantly decreases the size of the computational domain and increases computational performance. We use this capability in the second mesh compared, which

is the same as the QU60km mesh but with equatorial cells removed resulting in a mesh with 33,070 cell centers and 69,482 vertices, ∼29% and ∼30% of original cell centers and vertices in the global QU60km mesh. To determine which cells to keep and which to remove, simulations with the full mesh are used to determine a mask of cells where sea ice had ever existed during those simulations. These cells, as well as a buffer region of surrounding cells extending 1000km further, are kept in the reduced mesh. Simulations are performed on the *Anvil* machine at the Laboratory Computing Resource Center at Argonne

National Laboratory, which consists of 240 nodes with 36 cores per node.

Load balancing is an issue with sea-ice modeling since the presence of sea-ice in high latitudes requires more computation for cells located in these regions than in equatorial regions. If the computational domain is partitioned without taking this into account, processors computing high latitude cells will take longer to compute a time step than processors containing equatorial cells, which will have to wait periodically for high latitude processors to catch up. This waiting time is wasted and contributes to

poor performance. We have implemented three methods to deal with this issue. The first is culling equatorial cells as discussed





above. In addition, two improved methods for partitioning the domain across processors have been developed. The standard domain partitioning method for MPAS uses the *metis* tool (Karypis and Kumar, 1999) to evenly divide the domain amongst processors without taking into account load balancing (labelled "*Simple*" in Fig. 22). The two improved domain partitioning methods aim to give fewer cells and work to processors computing high-latitude regions. The first improved partitioning method

(labelled "*Weight*" in Fig. 22) adds a weight to cells in the partition. *metis* then aims, during the partitioning, to set the sum of weights in each partition equal. Giving high latitude cells a larger weight than equatorial cells means fewer cells are included in high latitude partitions, improving load balancing. *metis* requires this weight to be an integer, which we set to the nearest integer to $(1 + fn)$, where $f$ is the fraction of the time the cell in question contains sea-ice from a previous simulation. We find that a value of 4 for $n$ maximizes performance. The second improved partitioning method (labelled "*Region*" in Fig. 22)

divides the globe into a polar and equatorial region based on the ice presence mask derived from previous simulations. These two regions are individually partitioned and each processor's domain consists of one partition from the equatorial region and one from the polar region. A processor's computational domain then consists of two discontiguous regions, one polar and one equatorial.

Figure 22 shows slightly sub-linear strong-scaling performance for MPAS-Seaice for both meshes and for the *Region* and

*Weight* partitioning methods below around 400 nodes. Above about 400 nodes the computational cost of exchanging halo information between processors begins to dominate and linear scaling is no longer expected. The *Simple* partition method also mostly shows near linear scaling except at around ten processors when load balancing issues begin to affect performance relative to the other partition methods. As expected, removing the equatorial cells reduces this effect. Both the *Region* and *Weight* partitioning methods improve load balancing by around the same amount. Choice of equatorial mesh culling and

partition method can affect performance by up to a factor of around four. The *Region* partition method seems to underperform at high processor number once we reach the limit of strong-scaling. This was found to be caused by a deficiency in the generated partitions where, at high processor number, boundaries between partitions would become tortuous.

Comparison of computational performance between MPAS-Seaice and CICE is non-trivial. While MPAS-Seaice can be used with quadrilateral meshes, its primary use is expected to be with SCVT meshes consisting primarily of hexagonal cells.

The ratio of velocity points to cell centers for these meshes is approximately two, whereas the quadrilateral grids used by CICE have approximately the same number of velocity points as cell centers. Comparing performance of MPAS-Seaice using SCVT meshes to CICE using quadrilateral meshes then depends on whether the performance is compared based on the number of cell centers or velocity points. To give an approximate idea of the relative performance of MPAS-Seaice and CICE we compare the SYPD achieved on *Anvil* for ten days of the simulation described in section 3.4 with both MPAS-Seaice and CICE

using a fully-global one degree quadrilateral grid. Simulation throughput for MPAS-Seaice and CICE is listed in Table 1 as simulated years per day (SYPD) for the whole model, and for the velocity solver, transport, and column schemes. For the total model MPAS-Seaice achieved approximately 70% of the CICE throughput. The better computational performance of CICE is expected since the unstructured mesh in MPAS-Seaice necessitates less efficient memory access patterns. As a percentage of CICE model performance, the MPAS-Seaice velocity solver displayed the poorest throughput, and the column physics the

most competitive throughput.





**Table 1.** Model throughput (in SYPD) for various code sections for MPAS-Seaice and CICE. Results from 32 processors on *Anvil* with MPAS-Seaice using the 'Region' partitioning method and CICE using the 'slenderX2' partitioning method.

| Model timer | MPAS-SI | CICE | MPAS/CICE (%) |
|---|---|---|---|
| Total | 46.6 | 66.7 | 70.0 |
| Velocity solver | 227 | 389 | 58.4 |
| Transport | 139 | 194 | 72.1 |
| Column | 110 | 140 | 78.9 |

## 5  Conclusions

We have described a new sea-ice model, MPAS-Seaice, and successfully validated the velocity solver and transport schemes in idealized test cases, on both planar and spherical grids. These schemes are closely based on those implemented on the quadrilateral grid used in the CICE sea ice model, but adapted for the polygonal cells of MPAS meshes. When using the variational scheme with Wachspress basis functions and a quadrilateral MPAS mesh, the velocity solver of MPAS-Seaice replicates the velocity solver algorithm of CICE, allowing rapid testing and validation.

We developed several other schemes for the strain rate and stress divergence spatial operators to compare with the variational Wachspress scheme. We find that, while the variational scheme, with the alternate area denominator formulation, has excellent error characteristics for the stress divergence operator, the one-sided stencil of the variational strain rate operators results in poor error characteristics. The weak scheme shows the opposite, with good error characteristics for the strain rate operators, but asymmetric integrals around the dual triangles of the SCVT mesh result in larger errors for the weak stress divergence operator. This suggests that the variational scheme could be improved by modifying the its strain rate operator to have a two sided stencil. We investigated several averaging techniques to implement a two-sided stencil for the variational scheme, which resulted in improved error characteristics for the variational strain rate operator. For basin-scale sea-ice simulations, however, these alternate operators had only a small effect on simulation results.

MPAS-Seaice and CICE share the sophisticated suite of column physics and BGC originally developed in CICE, again allowing the rapid development of MPAS-Seaice. Global simulations with realistic forcing have validated MPAS-Seaice against similar simulations with CICE and against observations for sea-ice concentration, extent and volume. MPAS-Seaice has been coupled into the Energy Exascale Earth System Model (Golaz et al., 2019; Burrows et al., 2020), and the validation experiments described here give confidence in the sea-ice results from E3SM simulations. MPAS-Seaice shows nearly linear strong-scaling performance, and the flexibility in mesh partitioning afforded by its use of an unstructured mesh allows efficient load balancing.

Future work will assess the fidelity and performance of MPAS-Seaice on variable-resolution meshes, and examine more recent metrics for evaluating sea-ice dynamics, such as new statistical metrics of linear kinematic features (e.g. Hutter and Losch, 2020). Potential challenges with variable resolution meshes include assessing the resolution invariance of sea-ice rhe-





ologies and developing resolution-invariant versions, efficiently using future heterogeneous computing architectures, as well as generating efficient domain partitions of highly variable-resolution meshes.

*Code and data availability.* MPAS-Seaice v1.0.0 is released as part of the Energy Exascale Earth System Model (E3SM) version 2, and is available at https://doi.org/10.11578/E3SM/dc.20210927.1.

*Author contributions.* AKT developed and implemented the velocity solver and performed all simulations and analysis. WHL developed
and implemented the transport scheme. ECH, AKT and NJ packaged the thermodynamics into the Icepack library used by MPAS-Seaice. DE provided support for the analysis of the horizontal operators. TDR contributed to the development of the velocity solver. JW provided software support. AKT prepared the manuscript with contributions from all co-authors.

*Competing interests.* The authors declare that they have no conflict of interest.

*Acknowledgements.* This research was supported as part of the Energy Exascale Earth System Model (E3SM) project, funded by the U.S.
Department of Energy, Office of Science, Office of Biological and Environmental Research. The National Center for Atmospheric Research is a major facility sponsored by the National Science Foundation under Cooperative Agreement No. 1852977. This research used a high-performance computing cluster provided by the BER Earth System Modeling program and operated by the Laboratory Computing Resource Center at Argonne National Laboratory. The authors would like to thank Michael Duda of the National Center for Atmospheric Research for his contributions to the MPAS software infrastructure, and Sara Calandrini (LANL) for help with mesh generation.



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
