# Peer review of "MPAS-Seaice (v1.0.0): Sea-ice dynamics on unstructured Voronoi meshes"

_Geoscientific Model Development, 2021_

## Referee Comment (RC1)

**Review of** 'MPAS-Seaice (v1.0.0): Sea-ice dynamics on unstructured Voronoi meshes' by A. K. Turner et al.

I can only recommend this manuscript for the journal. It describes the MPAS-Seaice model which is based on the B-grid discretization for meshes obtained by the Spherical Centroidal Voronoi Tesselation. The sea ice velocity is placed at the corners of mesh polygons (generally hexagons). This leads to a discretization that is similar to that using sea ice velocities placed on triangle centers on triangular grids, which have been proposed earlier. However, MPAS-Seaice uses different methods (variational and weak) to compute the strain rates and the divergence of stresses. The description of these methods presents the most interesting part of the manuscript (in my opinion). The incremental remapping scheme used for the advection of scalars follows essentially the earlier paper, but even in this case there are details that might be of interest to the community.

The difficulty of B-grid discretization on quasi-hexagonal meshes lies in the existence of a numerical mode in velocities. This mode is related to the fact that the two corners of hexagons sharing a common edge (or the centers of triangles on a dual mesh) are geometrically different. One can see this in a transparent way by looking at triangles of dual mesh: The orientation of the pattern of neighbors is different for two triangles sharing a dual edge. This leads to some difference in discrete differential operators for these two triangles, hence the mode in velocities. This mode can be filtered by the discretization, which is the case for the weak method, and this mode can be emphasized by the discretization, which is the case for the variational method. While filtering may lead to better accuracy for strain rates, it leaves a non-trivial kernel in strain rates, i.e., there are non-trivial discrete sea ice velocities, which lead to zero stress divergence and thus may contaminate solutions. The variational method might show less accurate strain rates because of the the contribution of spurious velocity mode, but the mode will be suppressed in dynamics, and no null space will be created. The manuscript does not consider these details, and I do not think it should. However, they might be helpful in interpreting the results on the accuracy of strain rate and stress divergence computations. I think that the availability of variational and weak methods is an advantage of MPAS-Seaice, and future practice will show which one is optimal. I tried the weak method in FESOM (slightly modified for median-dual control volumes of FESOM2), but was satisfied with its performance only when I added stabilization that removes the null-space. So my guess is that the variational method will be more reliable.

In summary, this is an excellent contribution which should be published.

Sergey Danilov

**Minor comments**

line 26 'parallelization' does not require the EVP, but the EVP may improve

scalability.

28-29 Why metric terms are specially mentioned? They must be present.

71 'twelve pentagons' are misleading if we think about the ocean mesh which does not cover the entire sphere.

85 'we rotate u and v' – the coordinate system is rotated, not u and v.

113-114 'Here, the directions ...' This has already been said above.

115 Say that you ignore boundary effects for simplicity, they simply lead to additional terms on the rhs of (3).

119 → 'contributions'

Formula (4) Already here the readers should be informed that the values of stresses are considered as parameters in $D$. It would be better to mention this explicitly in the argument list. The formula is misleading otherwise, for sigma is also a function of velocities. A still better approach is to take a test function instead of $\mathbf{u}$, as one will do in the finite element method.

179 'methods'→ basis functions

186 Does it imply that $2n_v^2$ coefficients have to be stored per scalar cell?

210 In FESOM sea ice implementation we define metric cosine at triangles, compute the areas of triangles, and then compute the areas of median-dual control volumes by summing $1/3$ of the contributions from triangles. This is the same as taking the mass matrix and lumping it. In this way our scalar areas are always consistent with the areas of elements where the cosine of latitude is estimated. Your (34) is the lumped mass matrix too. In reality, if one thinks about your discretization as an example of finite-element method, the question about the control volumes associated to velocity points does not occur (which means that they appear as the result of the method). Let us write $\partial_t \mathbf{u} = \nabla \cdot \boldsymbol{\sigma}$. Multiply this with a test function $\mathbf{w}$, integrate, and apply the integration by parts in the rhs:

$$\partial_t \int \mathbf{w} \cdot \mathbf{u} dA = - \int \nabla \mathbf{w} : \boldsymbol{\sigma} dA,$$

where the boundary terms are omitted. The rhs here is the same as in (3) of the manuscript if one takes into account that $\boldsymbol{\sigma}$ is symmetric tensor. Now take $\mathbf{w} = \mathbf{w}_j W_j$, $\mathbf{u} = \mathbf{u}_j W_j$ where summation is implied over the repeating indices. The result is

$$\mathbf{w}_i M_{ij} \partial_t \mathbf{u}_j = - \int \nabla (\mathbf{w}_i W_i) \boldsymbol{\sigma} dA,$$

where $M_{ij} = \int W_i W_j dA$ is the mass matrix. The requirement that the equation above holds for arbitrary $\mathbf{w}_j$ is equivalent (for the rhs) to differencing of $D$ over $u_i$ and $v_i$ (take $\mathbf{w}_i = (1,0)$ or $(0,1)$). Thus we obtain the equations which correspond to your (32). Lumping the mass matrix leads to your (34). Thus, (34) gives the only consistent definition for the area of dual cells. Other definitions are not allowed.

228 The variational method is also a 'weak' method, differential operators are obtained by projecting on some functions. Your 'weak' method is actually

a finite-volume method.

Equation (38): one can start from it from the very beginning. Since the metric terms are added independently, it is sufficient to know how to express a gradient of scalar u and v, which is just the generalized divergence theorem.

Equation (46): Here, a part of metric terms is taken twice. Look at (45). If one takes a vector quantity instead of the tensor sigma, (45) will be an exact finite volume divergence. No metric terms are needed, they all are taken into account through $l_i$. In the case of second rank tensor, $\boldsymbol{\sigma}_i \cdot \mathbf{n}_i$ is a vector, and unit vectors are different at different $i$. Bringing them to a single location is equivalent to the account of the contributions from differentiating unit vectors. The result will be $(-\sigma_{12}, \sigma_{11}) \tan\theta / r$ in (46). Once again, the reason lies in using (45) for the primed quantities. Had you used finite differences to compute the divergence, the metric terms in (46) would be correct.

444 Why do you need a citation here? This is so by definition.

446 I doubt that it is of interest to look at the behavior of gradients at a point, because only weighted combination of gradients appears in the code. Averaged gradients make much more sense.

475 Indeed, the operator of gradient or divergence on triangle is only first order accurate.

A general comment here: If one starts from a smooth $f$ sampled at the corners of primary mesh, there might be a grid-scale noise in the divergence of stresses for the variational method. The weak method may show cleaner results because of using a larger stencil for strain rate computations. However, conclusions can be different for sea ice dynamics evolving with time. The noise will likely be suppressed by dynamics in the variational method. In contrast, for the weak discretization, if some grid-scale noise will be added to the velocity, it may stay longer if it does not affect the discrete stress divergence. I think the behavior in 3.1.1 and 3.1.2 can be partly explained through the presence of mode. The transient behavior can be of interest as well (but not for this paper).

705 '400 nodes' Do you mean cores? Also it would be of interest to present the results shown in Fig. 22 in terms of the mean number of mesh cells per core. This should make the results for global and polar meshes similar, and the statement on the scaling will be more general. In FESOM (see www.geosci-model-dev.net/12/3991/2019/) scaling of sea ice is always sublinear, but continues to about 200-300 vertices (equivalent to cells on the Voronoi mesh) per core, which is similar to your result.

---

## Author Comment (AC1)

**Response to reviewers of "MPAS-Seaice (v1.0.0): Sea-ice dynamics on unstructured Voronoi meshes"**

Adrian K. Turner, William H. Lipscomb, Elizabeth C. Hunke, Douglas W. Jacobsen, Nicole Jeffery, Darren Engwirda, Todd D. Ringler, and Jonathan D. Wolfe

February 1, 2022

The authors would like to thank all the reviewer of this paper for their insightful comments and suggested corrections.

**1 Reviewer 1**

I can only recommend this manuscript for the journal. It describes the MPAS-Seaice model which is based on the B-grid discretization for meshes obtained by the Spherical Centroidal Voronoi Tesselation. The sea ice velocity is placed at the corners of mesh polygons (generally hexagons). This leads to a discretization that is similar to that using sea ice velocities placed on triangle centers on triangular grids, which have been proposed earlier. However, MPAS-Seaice uses different methods (variational and weak) to compute the strain rates and the divergence of stresses. The description of these methods presents the most interesting part of the manuscript (in my opinion). The incremental remapping scheme used for the advection of scalars follows essentially the earlier paper, but even in this case there are details that might be of interest to the community.

The difficulty of B-grid discretization on quasi-hexagonal meshes lies in the existence of a numerical mode in velocities. This mode is related to the fact that the two corners of hexagons sharing a common edge (or the centers of triangles on a dual mesh) are geometrically different. One can see this in a transparent way by looking at triangles of dual mesh: The orientation of the pattern of neighbors is different for two triangles sharing a dual edge. This leads to some difference in discrete differential operators for these two triangles, hence the mode in velocities. This mode can be filtered by the discretization, which is the case for the weak method, and this mode can be emphasized by the discretization, which is the case for the variational method. While filtering may lead to better accuracy for strain rates, it leaves a non-trivial kernel in strain rates, i.e., there are non-trivial discrete sea ice velocities, which lead to zero stress divergence and thus may contaminate solutions. The variational method might show less accurate strain rates because of the the contribution of spurious velocity mode, but the mode will be suppressed in dynamics, and no null space will be created. The manuscript does not consider these details, and I do not think it should. However, they might be helpful in interpreting the results on the accuracy of strain rate and stress divergence computations. I think that the availability of variational and weak methods is an advantage of MPAS-Seaice, and future practice will show which one is optimal. I tried the weak method in FESOM (slightly modified for median-dual control volumes of FESOM2), but was satisfied with its performance only when I added stabilization that removes the null-space. So my guess is that the variational method will be more reliable.

In summary, this is an excellent contribution which should be published. Sergey Danilov

- We thank the reviewer for the insight into issues with the B-grid formulation. This will be very useful during our forthcoming investigation of MPAS-Seaice on variable resolution meshes.

**1.1 Minor comments**

• line 26 'parallelization' does not require the EVP, but the EVP may improve scalability.

- Modified the line to read: "to allow explicit time-stepping and improved parallelization scalability of the algorithm."
- 28-29 Why metric terms are specially mentioned? They must be present.
  - Removed explicit mention of metric terms here.
- 71 'twelve pentagons' are misleading if we think about the ocean mesh which does not cover the entire sphere.
  - Modified the text to read: "but at least twelve pentagons are needed to complete the tessellation when cells cover the entire sphere. Less pentagons would be required for an ocean/sea-ice mesh where only part of the sphere is covered in cells."
- 85 'we rotate u and v' the coordinate system is rotated, not u and v.
  - Modified the line to read: "Consequently, we rotate the coordinate system so that the pole of u and v lies on the geographical equator at 0° longitude."
- 113-114 'Here, the directions ...' This has already been said above.
  - This line has been removed.
- 115 Say that you ignore boundary effects for simplicity, they simply lead to additional terms on the rhs of (3).
  - We remove the reference to boundary effects before the equation and after the equation add the line: "For simplicity we ignores boundary effects, which would add additional terms to the right-hand side of this equation."
- $119 \rightarrow$  'contributions'
  - Changed "contribution" to "contributions".
- Formula (4) Already here the readers should be informed that the values of stresses are considered as parameters in D. It would be better to mention this explicitly in the argument list. The formula is misleading otherwise, for sigma is also a function of velocities. A still better approach is to take a test function instead of **u**, as one will do in the finite element method.
  - We have added the stresses as parameters to the functional. We have also expanded the variational part of the derivation. While it may seem over explicit now, we believe this explanation will be useful to readers unfamiliar with this variational methods and finite element methods in general.
- 179 'methods'  $\rightarrow$  basis functions
  - Changed "methods" to "basis functions"
- 186 Does it imply that  $2n_v^2$  coefficients have to be stored per scalar cell?
  - We add the line "For the variational method pre-computed values for the variables  $\mathcal{S}_{lm}^x, \mathcal{S}_{lm}^y, \mathcal{T}_{lm}, \frac{\partial \mathcal{W}_m}{\partial x}\Big|_l$ , and  $\frac{\partial \mathcal{W}_m}{\partial y}\Big|_l$  must be stored, resulting in a total of  $5n_v^2$  values stored per cell." to clarify this point.
- 210 In FESOM sea ice implementation we define metric cosine at triangles, compute the areas of triangles, and then compute the areas of median-dual control volumes by summing 1/3 of the contributions from triangles. This is the same as taking the mass matrix and lumping it. In this way our scalar areas are always consistent with the areas of elements where the cosine of latitude is estimated. Your (34) is the lumped mass matrix too. In reality, if one thinks about your discretization as an example of finite-element method, the question about the control volumes associated to velocity points does not

occur (which means that they appear as the result of the method). Let us write  $\partial_t \mathbf{u} = \nabla \cdot \sigma$ . Multiply this with a test function  $\mathbf{w}$ , integrate, and apply the integration by parts in the rhs:

$$\partial_t \int \mathbf{w} \cdot \mathbf{u} dA = -\int \nabla \mathbf{w} : \sigma dA,\tag{1}$$

where the boundary terms are omitted. The rhs here is the same as in (3) of the manuscript if one takes into account that  $\sigma$  is symmetric tensor. Now take  $\mathbf{w} = \mathbf{w}_j W_j$ ,  $\mathbf{u} = \mathbf{u}_j W_j$  where summation is implied over the repeating indices. The result is

$$\mathbf{w}_i M_{ij} \partial_t \mathbf{u}_j = -\int \nabla(\mathbf{w}_i W_i) \sigma dA,\tag{2}$$

where  $M_{ij} = \int W_i W_j dA$  is the mass matrix. The requirement that the equation above holds for arbitrary  $\mathbf{w}_j$  is equivalent (for the rhs) to differencing of D over  $u_i$  and  $v_i$  (take  $\mathbf{w}_i = (1, 0)$  or (0,1)). Thus we obtain the equations which correspond to your (32). Lumping the mass matrix leads to your (34). Thus, (34) gives the only consistent definition for the area of dual cells. Other definitions are not allowed.

- We thank the reviewer for the explanation. We add the following text: "This is the equivalent of lumping the mass matrix in the finite element method (Reddy, 1993)."
- 228 The variational method is also a 'weak' method, differential operators are obtained by projecting on some functions. Your 'weak' method is actually a finite-volume method.
  - Renamed the weak method to "finite-volume" method.
- Equation (38): one can start from it from the very beginning. Since the metric terms are added independently, it is sufficient to know how to express a gradient of scalar u and v, which is just the generalized divergence theorem.
  - We change the text to start from equation 38 as suggested.
- Equation (46): Here, a part of metric terms is taken twice. Look at (45). If one takes a vector quantity instead of the tensor sigma, (45) will be an exact finite volume divergence. No metric terms are needed, they all are taken into account through  $l_i$ . In the case of second rank tensor,  $\sigma_i \cdot \mathbf{n}_i$  is a vector, and unit vectors are different at different *i*. Bringing them to a single location is equivalent to the account of the contributions from differentiating unit vectors. The result will be  $(-\sigma_{12}, \sigma_{11}) \tan \theta/r$  in (46). Once again, the reason lies in using (45) for the primed quantities. Had you used finite differences to compute the divergence, the metric terms in (46) would be correct.
  - Changed the metric definition and changed the text to the following: "Metric terms in the divergence of a second-order tensor, like the stress tensor, have two contributions: the first comes from the varying grid cell size and the second from the varying directions of the coordinate basis vectors (Malvern, 1969). Eq. (42) accounts for the first of these effects, and in order to account for the second effect the stress divergence becomes:

$$(\nabla \cdot \sigma)_u = (\nabla \cdot \sigma)'_u - \frac{\sigma_{12} \tan \theta}{r}$$

$$(\nabla \cdot \sigma)_v = (\nabla \cdot \sigma)'_v + \frac{\sigma_{11} \tan \theta}{r}$$
(3)

where the components of  $\sigma$  are approximated as the average of the values on the dual mesh vertices.

- 444 Why do you need a citation here? This is so by definition.
  - Removed the reference.

- 446 I doubt that it is of interest to look at the behavior of gradients at a point, because only weighted combination of gradients appears in the code. Averaged gradients make much more sense.
  - We modified the scaling diagrams to use area integrals for variational derivatives/strains and line integrals for the finite volume derivatives and strains. Added the following text: "For the spatial derivatives examined in this section and the strain rates examined in section 3.1.2, we calculate the  $L_2$  norm with the following methods. For the variational operators we integrate the square of the error across the domain using the variational basis functions defined in section 2.2.1 for the quantity of interest. The  $L_2$  norms for the variational derivatives and strains are given by

$$L_2 = \sqrt{\frac{\sum_i \int_i (\sum_l \mathcal{W}_l f_{il} - \hat{f}(x, y))^2 dA}{\sum_i \int_i \hat{f}(x, y)^2 dA}}$$
(4)

where the sum over i is a sum over cells in the region of interest, and the area integral is over cell i, performed by splitting the polygon into sub-triangles and using the 8th order integration rules of Dunavant (1985). The modeled quantity of interest is determined within the interior of the cell from the basis functions,  $W_l$ , and quantity of interest,  $f_{il}$ , on vertex l of cell i. For derivatives/strains calculated with the Wachspress basis function we perform the integration with the Wachspress basis functions and like-wise for the PWL basis functions.  $\hat{f}(x, y)$  are the analytical values of the field of interest within the cell. For the finite-volume derivative/strain operators we perform line integrals of the square of the error around the dual mesh cell surrounding primary cells points. The  $L_2$  norms for the finite-volume derivatives and strains are given by

$$L_{2} = \sqrt{\frac{\sum_{i} \sum_{j} l_{j} \int_{0}^{1} (f_{j}(\chi) - \hat{f}_{j}(\chi))^{2} d\chi}{\sum_{i} \sum_{j} l_{j} \int_{0}^{1} \hat{f}_{j}(\chi)^{2} d\chi}}$$
(5)

where the sum *i* is over the primary cells, and the sum *j* is over the edges of the dual cell surrounding cell *i*. Coordinate  $\chi$  signifies the position along edge *j*, with  $f_j(\chi)$  calculated from a linear interpolation from the derivative/strain values at each end of the edge.  $\hat{f}_j(\chi)$  are the analytical values of the field of interest along the edge. The line integrals are performed with 7th order Gauss-Lobatto quadrature rules. These formulas emulate how the stress values, derived from the derivative/strain values, are used in their respective stress divergence operators." and "For the  $L_2$  error norm calculated for the stress divergence operators in this section and in section 3.1.2, we use

$$L_{2} = \sqrt{\frac{\sum A_{i}(f_{i} - \hat{f}_{i})^{2}}{\sum A_{i}\hat{f}_{i}^{2}}}$$
(6)

where the sum is over either grid cells, or vertices in the region of interest,  $A_i$  is the area of either the primary cell, or the dual cell surrounding the vertex, and  $f_i$  and  $\hat{f}_i$  are the model and analytical values of the field of interest, respectively."

- 475 Indeed, the operator of gradient or divergence on triangle is only first order accurate. A general comment here: If one starts from a smooth f sampled at the corners of primary mesh, there might be a grid-scale noise in the divergence of stresses for the variational method. The weak method may show cleaner results because of using a larger stencil for strain rate computations. However, conclusions can be different for sea ice dynamics evolving with time. The noise will likely be suppressed by dynamics in the variational method. In contrast, for the weak discretization, if some grid-scale noise will be added to the velocity, it may stay longer if it does not affect the discrete stress divergence. I think the behavior in 3.1.1 and 3.1.2 can be partly explained through the presence of mode. The transient behavior can be of interest as well (but not for this paper).
  - We thank the reviewer for the insight of this comment. We plan to explore these issues more fully in our forthcoming investigation of MPAS-Seaice on variable resolution meshes.

- 705 '400 nodes' Do you mean cores? Also it would be of interest to present the results shown in Fig. 22 in terms of the mean number of mesh cells per core. This should make the results for global and polar meshes similar, and the statement on the scaling will be more general. In FESOM (see www.geoscimodel-dev.net/12/3991/2019/) scaling of sea ice is always sublinear, but continues to about 200-300 vertices (equivalent to cells on the Voronoi mesh) per core, which is similar to your result.
  - "Nodes" changed to "cores". Added a second subpanel to figure 22 with the performance data plotted against cells per processor. Added the text: "Figure 22b shows the same performance results plotted against the average number of cells per core. Sub-linear scaling is found when the number of cells per core is greater than  $\sim 300$ , with a degradation in performance for simulations with fewer cells per core than this value. A similar result was found for FESOM (Koldunov et al. 2019)."

**2 Reviewer 2**

This paper describes a new sea-ice model (MPAS-Seaice) that uses a Spherical Centroidal Voronoi Tesselation (SCVT) unstructured mesh. The model-calculated velocities, strain rates, and stress divergence terms are compared to those from the CICE model in idealized test cases using three different methods for representing quantities on the grid: Wachspress basis functions, piecewise linear basis functions, and the "weak" (integral) method. Global simulations are run with realistic forcing, and the resulting sea-ice extent, thickness, and volume are compared to CICE and to observations. MPAS-Seaice closely reproduces the results of CICE when a quadrilateral mesh is specified. Although MPAS-Seaice runs about 30% slower than CICE, the SCVT mesh gives it greater flexibility to provide higher resolution in selected areas. MPAS-Seaice is the current sea-ice component of the E3SM global climate model.

This is an excellent paper – thorough, well organized, and clearly written. It provides a good description and a convincing validation of the MPAS-Seaice model. It could almost be published in present form, save for a few minor technical corrections and suggestions.

**2.1 Minor technical corrections and suggestions**

- Line 15. MPAS "runs 70% as fast as CICE". I honestly had trouble figuring out if this means MPAS is faster or slower than CICE. If MPAS runs in 70% of the time as CICE then it's faster than CICE. But actually MPAS is slower (Table 1). Consider writing something like: "runs 30% slower than CICE" or "runs with 70% of the throughput of CICE"
  - Changed this line to "runs with 70% of the throughput of CICE".
- Lines 15-16. "culling of equatorial model cells". I know what culling means, but I had trouble understanding it in this context until I got to page 38, where I found out that it simply means REMOVING, as stated on lines 677 and 680. To me, the word "cull" has the connotation of selective removal due to some deficiency, such as culling sickly animals from a herd. What you really mean is "excise" – to cut out or remove. I suggest using "remove" throughout the text rather than "cull" (lines 15, 634, 655, Figure 22 caption, 690, 709).
  - Changed "cull" to "remove" throughout.
- Line 23. Does "CICE" stand for something? If yes, spell it out on first usage.
  - No, CICE is not an acronym.
- Good Introduction.
  - Thanks!
- Line 69 and Figure 1. I'm just curious, are the edge points midway between the cell centers? If yes, you could mention it.

- Added ", with the edge equidistant from the two cell centers."
- Line 120. "consists" should be "consist"
  - Changed "consists" to "consist".
- Equation (4). On the right-hand side, in the list of arguments of the function D, the variable "un" should be "un". Same comment for equations (5), (6), (7), (8), (32), (33).
  - This has been corrected through out.
- Equation (6). On the left-hand side, the cell area is denoted " $A_{ui}$ ". Why the subscript "u"? Couldn't the area just be denoted " $A_i$ "?
  - Changed " $A_{ui}$ " to " $A_i$ ".
- Equation (19). It is misleading to write "=0" at the end of this equation. If  $L_j(x, y) = 0$  then according to equation (17)  $N_i$  is also zero.  $L_j(x, y)$  is only zero along the *j*-th edge of the polygon, not for all (x, y). I suggest writing something like this:

 $L_j(x,y) = 1 - a_j x - b_j y$  (19)

where  $a_j$  and  $b_j$  are defined by the condition that Lj(x,y) = 0 along the *j*-th edge of the polygon."

- Made this suggested change.

• Equation (21). Why not choose xc = the cell center as shown in Figure 1?

- The chosen formulation was the simplest implementation.

- Line 186. "can be calculated once FOR EACH CELL" (right?)
  - Added "for each cell" to the text.
- Equation (25). Delete the period (.) at the end.
  - Removed period.
- Figure 4. There are a lot of letters in this figure! To improve readability, I suggest putting a small black square at the location of each vertex, as in Figures 1 and 3.
  - Black squares added to vertex points and "Vertices are also shown with a square marker." added to figure caption.
- Line 364. V1 and V2 should be  $V_1$  and  $V_2$
  - Corrected V1 and V2 to  $V_1$  and  $V_2$
- Line 441.  $A_{di}$  should be  $A_i$
  - $A_{di}$  changed to  $A_i$
- Lines 461-462. It should be noted after this sentence that averaging the Weak method makes the error LARGER.
  - Added ", although this averaging increases the error relative to the original weak scheme." to the end of this sentence.
- Lines 469-470. "For the hexagonal cell mesh the variational methods show lower errors with better convergence rates than the weak method." I suggest deleting this sentence because it is out of place here, and it is repeated on lines 473-475 in slightly different form: "For the hexagonal cell mesh, the weak method shows only first-order convergence with significantly higher error than the variational method."

- We have deleted the sentence.
- Lines 480-481. The end of this sentence, which refers to the averaged Wachspress scheme, is true for Figure 5c but not for Figures 5a,b,d.
  - Modified the line to read: "In summary, for regular planar meshes, for the gradient operators, the weak and averaged PWL and averaged weak schemes show a higher order of error convergence and lower absolute errors than the variational schemes, while the averaged Wachspress scheme shows a higher order of convergence than the un-averaged Wachspress scheme, except for hexagonal cell meshes when the x derivative is being calculated.".
- Line 481. Wachspress is mis-spelled.
  - Spelling corrected.
- Lines 482-483. This sentence (which refers to Figure 6) is true for the hexagonal mesh (solid lines) but not for the square mesh (dashed lines).
  - Modified the line to read: "Conversely, for the stress divergence operator and hexagonal cell meshes, the variational schemes show lower absolute errors and better error convergence than the weak scheme, while for square cell meshes the order of convergence is the same between the variational and weak schemes with the weak scheme producing lower absolute errors.".
- Line 494. I think "strain rate operators" should be "strain rate component eps11"
  - This has been corrected.
- Line 507 or later. It should be noted in the discussion of Figure 9 that averaging the Weak method makes the error larger.
  - Added the line: "As for the hexagonal cell planar mesh, for both the  $L_2$  and  $L_{\infty}$  norms, averaging the weak scheme increases the error.".
- Figure 7. Why do the velocity components u and v have primes (') on them?
  - The primes signify that the u and v directions are the rotated ones, and not the actual eastwards and northwards directions.
- Figure 7 caption, line 2. "e-f" should be "e-g" or "e,f,g"
  - Corrected "e-f" to "e-g"
- Figure 9 caption, line 4. Add "as dotted lines" at the end of the sentence.
  - Added "as dotted lines".
- Figure 17 caption, line 2. Delete "a" before "short dashed"
  - Removed "a".
- Figure 19. To the right of panel (L), the label for the color scale should say "ice conc. diff." (not "ice conc.")
  - This has been fixed in the figure.
- Figure 20. To the right of panel (i), the label for the color scale should say "ice thickness diff." (not "ice thickness")
  - This has been fixed in the figure.

- Figure 22. The solid black curve has double tick-marks along its length, but this does not correspond with the legend. It looks to me like the tick-marks are not necessary on any of the curves: black denotes Global, red denotes Polar, and the line style (solid, dashed, dotted) denotes Simple, Region, or Weight. If symbols on the curves are desired, they could all be solid dots.
  - We removed markers from the line plots to increase clarity.
- Lines 677-684. In the discussion of removing equatorial cells, something about boundary conditions should be mentioned. On a global grid there are no boundaries, but when equatorial cells (or any group of cells) are removed, suddenly there are boundaries to deal with. How is that handled?
  - We added the line: "Removal of equatorial cells produces extra domain boundaries mid-ocean. These are treated as regular land-ocean boundaries, but because of the buffer region mentioned above, during physically reasonable simulations sea-ice will not encounter them.".
- Line 737. "the its" delete one or the other of these words.
  - Removed "the"
- Line 745. Note that "nearly linear" refers to log-log space.
  - Modified the text to read: "MPAS-Seaice shows power-law strong-scaling performance with a nearly linear exponent"

**3 Reviewer 3**

The article describes the sea ice model MPAS, which is used as a sea ice model in the climate model E3SM. A distinguishing feature of the model is the use of Voronoi grids. The ice velocities are specified in the nodes of the cells, the tracers in the midpoints. The grids can be thought of as dual grids to triangular grids, for example. However, MPAS also offers the possibility to use quadrilateral grids.

The main focus of the article is the discretization of the local differential operators for setting up the moment equation. Two different approaches are presented and discussed here. Later, numerical test calculations are also carried out.

In the further course, the authors describe an adapted advection scheme and discuss the embedding of the dynamical model into a climate model and present different numerical experiments.

The article is insteresting and clearly written and gives very good documentation of the aspects of discretization, technical details and clearly highlights the implications of different discretization approaches.

I clearly recommend acceptance of the paper for publication and have only a few comments. Some minor inaccuracies as well as typos have already been pointed out in other reviews.

Some remarks:

- (4), (5), (6), etc. Is there the index 'd' missing in the enumeration of  $u_1, u_2, \dots$ ?
  - There was a missing d from the  $u_{n_d}$  velocity component, which has been corrected. The discretized velocity components are labelled  $1, 2, 3, ..., n_d$  so no other d index is required.
- 148-150 and (13). This property of a basis is usually called "Lagrangian" or "Lagrange Form"
  - Changed text to read: "To perform this integral we use a set of Lagrangian basis functions"
- The EVP approach, using distinct variables for the stresses, is similar to a "mixed formulation" or approximation of of the VP model. For reasons of stability, an optimal choice would be to choose as function space for the stresses sigma the gradient of the velocity function space. At least in a standard finite element formulation, using the same order for both velocity and stresses would not be succesful since the inf-sup condition would be violated. Specifically I am for instance referring to (22), (23). This could the origing of some instabilities that later need to be any kind of stabilization, e.g. by a subsequent averaging.

- We thank the reviewer for this suggestion and plan to investigate these suggested changes to the basis functions in our forthcoming investigation of MPAS-Seaice on variable resolution meshes.
- I find it very difficult to discern the exact meaning and significance of Figure 4.
  - We have remade the figure, improving clarity and modified the caption adding "Backward trajectories are shown as red arrows, ending in departure points  $D_1$  and  $D_2$ . These backward trajectories define the departure region which is shaded in blue.".
- *l* 447-448: "This is understandable ..." what exactly do you mean here by the one-sided stencil and the "weak operator stencil surrounds ..."?
  - This is expanded to "This is understandable since the variational derivative operators only use velocity values from vertices on the same cell as the vertex that the derivative is being calculated for. These velocity values can then only occupy one half plane with respect to the derivative point, effectively creating a one sided stencil for the operator. Velocity values for the finitevolume operator, however, surround the derivative point, since here the derivative point is at the center not the side of the cell. This effectively creates a stencil surrounding the derivative location."
- In the summary, the authors do not sound as if they are completely convinced of their new model. Possibly this is just a realistic assessment, but the usual enthusiasm is missing when it comes to praising one's own work to the skies. Is there a clear winner when it comes to the two approaches "weak" and "variational"? Or does this essentially depend on the respective purpose. Major new developments were necessary in particular because the switch to unstructured grids was made. Here, too, I miss a final assessment: is this step worthwhile and can the advantages be used?
  - Our goal with the paper was to present an unbiased assessment of the methodology as possible. We do not believe there is a clear winner in the methodologies. As for the switch to unstructured meshes, as mentioned in the introduction, some coupling frameworks require ocean and sea ice to share meshes. The use of an unstructured ocean model then necessitates as unstructured sea ice model. Also, a prime benefit of unstructured meshes is the ability to have variable resolution meshes and concentrate computational resources in regions of interest - this aspect of the model will be assessed in future work.